



# 1 DynVarMIP: Assessing the Dynamics and Variability of

# 2 the Stratosphere-Troposphere System

Edwin P. Gerber[1] and Elisa Manzini[2]
[1] Courant Institute of Mathematical Sciences, New York University, 251 Mercer Street, New York NY
10012, USA.
[2] Max-Planck-Institut für Meteorologie, Bundesstraße 53, 20146 Hamburg, Germany
*Correspondence to:* Elisa Manzini  (elisa.manzini@mpimet.mpg.de)
**Abstract.** Diagnostics of atmospheric momentum and energy transport are needed to investigate the
origin of circulation biases in climate models and to understand the atmospheric response to natural and
anthropogenic forcing. Model biases in atmospheric dynamics are one of the factors that increase
uncertainty in projections of regional climate, precipitation, and extreme events. Here we define
requirements for diagnosing the atmospheric circulation and variability across temporal scales and for
evaluating the transport of mass, momentum and energy by dynamical processes in the context of the
Coupled Model Intercomparison Project Phase 6 (CMIP6). These diagnostics target the assessments of
both resolved and parameterized dynamical processes in climate models, a novelty for CMIP, and are
particularly vital for assessing the impact of the stratosphere on surface climate change.

**Keywords:** Atmosphere, dynamics, momentum and energy transfer, variability, climate and climate
change.
**1. Introduction**

The importance and challenge of addressing the atmospheric circulation response to global warming
have recently been highlighted by Shepherd (2014) and Vallis et al. (2015). Understanding circulation
changes in the atmosphere, particularly of the mid-latitude storm tracks, has been identified by the
World Climate Research Programme (WCRP) as one of the grand challenges in climate research. The
storm tracks depend critically on the transport of momentum, heat and chemical constituents
throughout the whole atmosphere. Changes in the storm tracks are thus significantly coupled with
lower atmosphere processes such as planetary boundary layer, surface temperature gradients and
moisture availability (e.g. Garfinkel et al., 2011, Booth et al., 2013) as well as with processes in the
stratosphere, from natural variability on synoptic to intraseasonal timescales (e.g. Baldwin and
Dunkerton, 2001) to the response to changes in stratospheric ozone (e.g. Son et al., 2008) and other
anthropogenic forcings (e.g. Scaife et al., 2012).

Rather then proposing new experiments, the strategy of the "Dynamics and Variability Model
Intercomparison Project" (DynVarMIP) is to request additional model output from standard CMIP
experiments. This additional output is critical for understanding the role of atmospheric dynamics in





past, present and future climate. Both resolved processes (e.g. Rossby waves) and parameterized
processes (e.g. gravity waves and the planetary boundary layer) play important roles in the dynamics
and circulation of the atmosphere in models.  DynVarMIP seeks to ensure that sufficient diagnostics of
all key processes in climate models are archived. Without this model output, we will not be able to
fully assess the dynamics of mass, momentum, and heat transport - essential ingredients in projected
circulation changes - nor take advantage of the increasingly accurate representation of the stratosphere
in coupled climate models. Our rational is that by simply extending the standard output relative to that
in CMIP5 for a selected set of experiments, there is potential for significantly expanding our research
capabilities in atmospheric dynamics.

Investigation of the impact of solar variability and volcanic eruptions on climate also relies heavily on
atmospheric wave forcing diagnostics, as well as radiative heating rates (particularly in the short wave).
By extending our request to the energy budget and including diagnostics such as diabatic heating from
cloud-precipitation processes, research on the links between moist processes and atmospheric dynamics
will be enabled as well. The interplay between moist processes and circulation is central to the WCRP
Grand Challenge on Clouds, Circulation and Climate Sensitivity (Bony et al., 2015).

The CMIP5 saw a significant upward expansion of models with a more fully resolved stratosphere (e.g.
Gerber et al., 2012), and several multi-model studies have investigated the role of the stratosphere in
present climate and in projections of future climate (e.g., Anstey et al., 2013; Charlton-Perez et al.,
2013; Gerber and Son, 2014; Hardiman et al. 2013; Lott et al., 2014; Manzini et al., 2014; Min and
Son, 2013; Shaw et al., 2014; Wilcox and Charlton-Perez, 2013) in addition to many other single
model studies. These studies document a growing interest in the role of middle and upper atmosphere
in climate (cf. Kidston et al., 2015). New research in this direction will take full advantage of the
DynVarMIP diagnostics.
**2. Objectives and Scientific Questions**

DynVar focuses on the interactions between atmospheric variability, dynamics and climate change,
with a particular emphasis on the two-way coupling between the troposphere and the stratosphere.  To
organize the scientific activity within the MIP, we have identified the following key questions:

•   How do dynamical processes contribute to persistent model biases in the mean state and
variability of the atmosphere, including biases in the position, strength, and statistics of the
storm tracks, blocking events, and the stratospheric polar vortex?
•   What is the role of dynamics in shaping the climate response to anthropogenic forcings (e.g.
global warming, ozone depletion) and how do dynamical processes contribute to uncertainty
in future climate projections and prediction?
•   How does the stratosphere affect climate variability at intra-seasonal, inter-annual and decadal
time scales?






Investigation of these topics will allow the scientific community to address the role of atmospheric
dynamics in the key CMIP6 science questions concerning the origin and consequences of systematic
model biases, the response of the Earth System to forcing, and how to assess climate change given
climate variability (Eyring et al this Special Issue). In particular, there is a targeted effort to contribute
to the storm track theme of the Clouds, Circulation and Climate Sensitivity Grand Challenge. The
DynVarMIP focus on daily fields and diagnostics of the atmospheric flow is also relevant to the Grand
Challenge on Climate Extremes, and could also enable contributions to the additional theme on
Biospheric Forcings and Feedbacks.
**3. The Diagnostics**

The DynVarMIP requests both enhanced archival of standard variables from the CMIP5 and new
diagnostics to enable analysis of both resolved and parameterized processes relevant to the dynamics of
the atmosphere. The diagnostics are organized around three scientific themes, as detailed below.

The diagnostics are requested from the DECK experiments, namely the AMIP atmosphere-only model
integrations [preferably for a minimum of 3 realizations] and selected 40-year periods of the
preindustrial control [years 111-150 after the branching point], abrupt4xCO2 [years 111-150] and
1pctCO2 [years 111-150] coupled model integrations. To allow comparisons with CMIP5, the
diagnostics are also requested for 40-year periods of the CMIP6 historical [1961-2000] and the
ScenarioMIP RCP8.5 [2061-2100] experiments (cf. Manzini et al. 2014). In addition, the DynVar
diagnostics (or relevant subsets thereof) are part of the diagnostic requests of AeroChemMIP, DAMIP,
DCPP, HighResMIP, and VolMIP [this Special Issue]. Note that modeling centers need only commit to
providing diagnostics to the DECK and the CMIP6 historical experiments, however, to participate in
the DynVarMIP.
**3.1 Atmospheric variability across scales (short name: *variability*)**

The first request of the DynVarMIP is enhanced archival of standard variables (listed in Table 1) as
daily and monthly means. While modeling centers have been archiving increasingly fine horizontal
resolution (close to the native model grid), vertical sampling has been limited to standard levels that
changed little from CMIP3 to 5.

The need for enhanced vertical resolution is particularly acute in the upper troposphere and lower
stratosphere (UTLS), where there are steep vertical gradients in dynamical variables (e.g. temperature
and wind) and chemical constituents (e.g. water vapor and ozone) across the tropopause. Without this
finer vertical resolution, analyses of the UTLS would be limited by vertical truncation errors,
preventing us from taking full advantage of increased horizontal resolution offered in new model
integrations.





A number of other MIPs, in particular HighResMIP (this Special Issue), have also recognized the need
for enhanced vertical resolution for daily data. A common proposed request, the "plev19" set of
pressure levels, has consequently been reached (Martin Juckes, personal communication, see:
https://earthsystemcog.org/site_media/projects/wip/CMIP6_pressure_levels.pdf). The pressure levels
of the plev19 set are 1000, 925, 850, 700, 600, 500, 400, 300, 250, 200, 150, 100, 70, 50, 30, 20, 10, 5,
and 1 hPa.
The diagnostics in Table 1 will allow for evaluation of atmospheric variability across time and spacial
scales, e.g. the assessment of model biases in blocking events, the tropospheric storm tracks, and the
stratospheric polar vortices. Comparison between the preindustrial control, historical, and idealized
(e.g. 1pctCO2 and RCP8.5) integrations will allow for evaluation of the response of atmospheric
variability to external forcings.
Novel to CMIP6 is also the daily zonal mean geopotential (zmzg, Table 1), tailored to the need of
DCPP (Decadal Climate Prediction Project) to analyze variability on longer time scales and for a large
number experiments, while minimizing storage requirements.
**3.2 Atmospheric zonal momentum transporialt (short name: *momentum*)**

The second group of diagnostics focuses on the transport and exchange of momentum within the
atmosphere and between the atmosphere and surface, and are listed in Tables 2, 3 and 4. Within this
group, a number of new (to CMIP) diagnostics and variables are requested. The goal of this set is to
properly evaluate the role of both the resolved circulation and the parameterized dynamical processes
in momentum transport. As daily timescales must be archived to capture the role of synoptic
processes, we focus on the zonal mean circulation, thereby greatly reducing the total output that must
be stored permanently. We have also prioritized the new variables, as noted in Tables 2, 3 and 4.
Priority 1 variables are essential to the MIP and required for participation. Priority 2 variables would
be very valuable to the MIP, but not are necessary for participation.
The zonal mean quantities are requested on the "plev39" vertical levels: 1000, 925, 850, 700, 600, 500,
146    400, 300, 250, 200, 170, 150, 130, 115, 100, 90, 80, 70, 50, 30, 20, 15, 10, 7, 5, 3, 2, 1.5, 1, 0.7, 0.5,
0.4, 0.3, 0.2, 0.15, 0.1, 0.07, 0.05, and 0.03 hPa. This fine sampling would allow for detailed
exploration of the vertical momentum transport. Subsampling is allowed for models with lower vertical
resolution or lower model tops.
Models largely resolve the planetary and synoptic scale processes that dominate the transport of
momentum within the free atmosphere. Quantification of this transport, however, depends critically on
vertical and horizontal wave propagation. The Transformed Eulerian Mean (TEM) framework allows
one to efficiently quantify this momentum transport by waves, in addition to estimating the Lagrangian
transport of mass by the circulation (e.g. Andrews and McIntyre, 1976; 1978). In the stratosphere, the
TEM circulation is thus far more relevant to transport of trace gases (e.g. ozone and water vapor) than





the standard Eulerian mean circulation (e.g. Butchart 2014). We have therefore request diagnostics
based on the TEM framework (see Table 2). The details of these calculations are presented in the
Appendix, and further insight can be found in the textbooks by Andrews et al., (1987; pages 127-130)
and Vallis (2006; chapter 12).

As seen in the Appendix, the TEM diagnostics depend critically on the vertical structure of the
circulation, i.e. vertical derivatives of basic atmospheric state and of wave fluxes. Even with the
enhanced "plev19" vertical resolution requested above, we would not be able to reproduce these
statistics from the archived output. It is therefore important that these calculations be performed on the
native grid of the model (or as close as possible), before being interpolated to standard levels for
archival purposes.

Dynamical processes, which need to be parameterized because they are not resolved on the grid of the
model, also play an important role in momentum transport. Gravity waves transport momentum from
the surface to the upper troposphere and beyond, but cannot be properly resolved at conventional GCM
resolution. Their wave stresses play a key role in the large scale circulation of the troposphere (e.g. the
storm tracks; Palmer et al., 1986) and are primary driver of the stratospheric circulation (e.g. Alexander
et al., 2010, and references therein). Atmospheric circulation changes have been shown to be sensitive
to the parameterization of gravity waves (e.g., Sigmond and Scinocca, 2010).  The availability of
tendencies from gravity wave processes (Table 2 and 3) will enable a systematic evaluation of this
driving term of the circulation, so far largely unexplored in a multi-model context.

Diagnostics to archive the parameterized surface stresses are listed in Table 4.  A number of studies
have documented that the large scale circulation and storm track structure are sensitive to the surface
drag (e.g. Chen et al. 2007; Garfinkel et al. 2011; Polichtchouk and Shepherd, in 2016). These
diagnostics will also allow us to connect the CMIP6 with the investigation of weather prediction
models by in the Working Group on Numerical Experimentation (WGNE) Drag Project
(http://collaboration.cmc.ec.gc.ca/science/rpn/drag_project/).  To understand how models arrive at the
total surface stress, we also request the component due to turbulent processes, usually parameterized by
the planetary boundary layer (PBL) scheme, including those stresses that come from subgrid
orographic roughness elements.  The role of other processes could then be diagnosed by residual.

Evaluation of the resolved and parameterized processes that effect the circulation are essential to
diagnosing and understanding persistent model biases in the mean state and variability of the
atmosphere.  In addition, a fundamental understanding of the underlying mechanisms driving the
response of the atmosphere to external forcing will improve confidence in future projections. We need
to know that models not only agree in the response, but that they agree for the same reasons.
**3.3 The atmospheric heat budget (short name: *heat*)**



This set of diagnostics allows us to understand the interaction between radiation, moisture, and the
circulation. As with our momentum diagnostics, we request only zonal mean statistics, to limit the
additional storage load (Table 5).
Breaking down the short and long wave heating tendencies is particularly important for understanding
the role of solar and volcanic forcing on the circulation. It will allow us to separate the direct impact of
changes in solar radiation and aerosol loading from the atmospheric response to these perturbations,
and enable analysis to break down feedbacks in Earth System models. Additional tendencies are
requested for gravity wave diagnostics, so that their contribution to the heat budget can be quantified
and compared.
**4. Analysis Plan**
DynVarMIP is holding a workshop in June 2016 to organize the exploitation of the requested
diagnostics.. The goal of the workshop is to coordinate analysis of the CMIP6 simulations, avoid
duplicate efforts, and ensure that our three scientific questions are investigated. At the June workshop,
we are planning to discuss and organize intermodel comparison papers to investigate the momentum
and heat balances of the historical climate (where it can be compared with observations and reanalysis),
and how model biases there relate to differences in the models's atmospheric circulation response to
external forcing, both in the idealized DECK perturbation experiments and in the RCP8.5. A follow up
workshop will be planned for 2018 or 2019 to ensure that scientific work continues forward.
The DynVarMIP has been based on our experience in coordinating community based, collaborative
analysis of coupled climate models from the CMIP5 through the SPARC DynVar activity (e.g. Gerber
et al., 2012). To enhance participation and collaboration with the modeling centers, representatives
have been invited to attend both the workshops and to participate in the scientific analysis and papers.
We have found that research on a mechanistic understanding of the atmosphere and on rectifying
model biases is often best organized organically, rather than from a top down approach. The TEM
diagnostics, for example, have been used in a number of CMIP5 studies (e.g. Hardiman et al., 2013;
Manzini et al., 2014), but had to be assembled on an ad hoc basis with a limited number models.
DynVarMIP is seeking to expand this research by making the key diagnostics available to all.
**5. Conclusions and Outlook**
The goal of the DynVarMIP is to evaluate and understand the role of dynamics in climate model biases
and in the response of the climate system to external forcing. This goal is motivated by the fact that
biases in the atmospheric circulation greatly limit our ability to project regional climate change, and
compromise our ability to project changes in extreme events.





Rather then proposing new experiments, DynVarMIP has organized a targeted list of variables and
diagnostics to characterize the role of both resolved and parameterized dynamical processes in the large
scale circulation of climate models. The DynVarMIP effort emerges from the needs of an international
community of scientists with strong connections to the modeling centers, with a long history (from the
SPARC/GRIPS workshops in the mid 1990s; Pawson et al., 2000). Given this participation, we expect
that the new diagnostics can be efficiently produced and will be fully utilized.

We are coordinating our efforts with several other MIPs. Transport plays a key role in the
AerChemMIP experiments with ozone depleting substances, making the TEM diagnostics particularly
relevant. The short-term VolMIP experiments and the DAMIP experiments focus in large part on
stratosphere-troposphere coupling, where the momentum and heat budget diagnostics are directly
relevant. Lastly, gravity wave effects and high frequency eddy processes are foci of the HiResMIP.
The availability of dynamically oriented diagnostics within the DECK and the CMIP6 historical will
provide the benchmark for these MIPs and others as well.

**Data availability:** The model output generated by the DynVarMIP diagnostic request will be
distributed through the Earth System Grid Federation (ESGF) with digital object identifiers (DOIs)
assigned. As in CMIP5, it will be freely accessible through data portals after registration. In order to
document CMIP6's scientific impact and enable ongoing support of CMIP, users are obligated to
acknowledge CMIP6, the participating modelling groups, and the ESGF centres. See Eyring et al (this
Special Issue) for further details.

**Appendix: TEM recipe**

This technical appendix outlines and gives recommendation on how to calculate the TEM diagnostics
for the momentum budget DynVarMIP output request (Table A1, subset of Table 2, section 3.2). For
the calculation of the TEM diagnostics we follow Andrews et al (1983, 1987). We recommend
calculating the diagnostics on pressure levels, on a grid very close or identical to that of the dynamical
core of the atmospheric model. For non-hydrostatic dynamical models in geometric-z coordinate, prior
to the diagnostic calculation it is necessary to transform the input variables to pressure coordinates, as
demonstrated by Hardiman et al (2010).

Given that the TEM diagnostics are usually displayed in a log-pressure vertical coordinate system (e.g.,
Butchart 2014), we thereafter detail how to transform the results to a standard log-pressure vertical
coordinate and so obtain the formulation of Andrews et at (1987), which is the one of our data request,
but for a re-scaling of the EP-flux.

*Coordinates, averages and frequency*



We recommend interpolating the fields of interest to pressure levels prior to taking zonal and temporal
averages (for both inline and offline calculations). Ideally, the pressure levels should be as close as
possible to the average position of the model levels, to minimize the impact of the interpolation.

Flux quantities with multiplying factors (e.g., heat flux v'θ') composed of anomalies from the zonal
mean (e.g., v' = v – zonal mean [v]) should be computed from high frequency data (6-hourly or higher
frequency) and their products then computed before averaging to daily or monthly mean.

Time averages are calculated by averaging over the day or month periods, either "offline" from model
outputs at 6-hour or higher frequency or directly computed over all time steps (i.e., "online").
Similarly, zonal averages are calculated averaging over all available longitudes, either offline (more
commonly) or online (seldom done).

***Input***

The input to the calculation of the TEM diagnostics, is given in Table A2. In the following to simplify
the writing of the TEM recipe, for the input we use:

$T$ for air temperature, ta variable in CMOR
$u$ for eastward wind velocity, ua variable in CMOR
$v$ for northward wind velocity, va variable in CMOR
$\omega$ for omega, wap variable in CMOR (vertical component of velocity in pressure coordinates, positive
down)
$p$ for pressure [Pa], plev dimension in CMOR
$\phi$ for latitude [radiant], derived from the latitude [degrees north] dimension in CMOR

Recommended constants for the calculation of the TEM diagnostics:

$p_0 = 101325$ Pa , surface pressure
$R = 287.058$ J K$^{-1}$kg$^{-1}$ , gas constant for dry air
$C_p = 1004.64$ J K$^{-1}$kg$^{-1}$ , specific heat for dry air, at constant pressure
$g_0 = 9.80665$ ms$^{-1}$ , global average of gravity at mean sea level
$a = 6.37123$ x $10^6$ m , earth's radius
$\Omega = 7.29212$ x $10^{-5}$ s$^{-1}$ , earth's rotation rate
$f = 2\Omega \sin \phi$, Coriolis parameter
$\pi = 3.14159$ , pi, mathematical constant

The following derivation of the TEM diagnostics makes use of the potential temperature, defined by:

$$\theta = T(p_0/p)^k$$

where $k = R/C_p$ is the ratio of the gas constant, $R$, to the specific heat, $C_p$, for dry air.






***TEM Diagnostics***

First, the input variables are zonally averaged and the anomalies from the respective zonally averaged
quantities are calculated. The zonally averaged quantities are denoted: $\bar{\theta}, \bar{u}, \bar{v}$ and $\bar{\omega}$. The anomalies:
$\theta', u', v'$ and $\omega'$.

Thereafter, fluxes and their zonal averages are calculated, for: $\overline{u'v'}$, the northward flux of eastward
momentum; $\overline{u'\omega'}$, the upward flux of eastward momentum; and $\overline{v'\theta'}$, the northward flux of potential
temperature.

Now we can proceed to calculate the Eliassen-Palm flux, **F**, its divergence, $\nabla \cdot \mathbf{F}$, the Transformed
Eulerian mean velocities, $\bar{v}^*$ and $\bar{\omega}^*$, the mass stream-function, $\Psi$.

The Eliassen-Palm flux is a 2-dimesional vector, $\mathbf{F} = \{F_{(\phi)}, F_{(p)}\}$, defined by:

$$F_{(\phi)} = a \cos \phi \left\{ \frac{\partial \bar{u}}{\partial p} \psi - \overline{u'v'} \right\} \text{, the northward component}$$
$$F_{(p)} = a \cos \phi \left\{ \left[ f - \frac{\partial \bar{u} \cos \phi}{a \cos \phi \partial \phi} \right] \psi - \overline{u'\omega'} \right\} \text{, the vertical component}$$

where: $\psi = \overline{v'\theta'} / \frac{\partial \bar{\theta}}{\partial p}$ is the eddy stream-function

The Eliassen-Palm divergence, $\nabla \cdot \mathbf{F}$, is defined by:

$$\nabla \cdot \mathbf{F} = \frac{\partial F_{(\phi)} \cos \phi}{a \cos \phi \, \partial \phi} + \frac{\partial F_{(p)}}{\partial p}$$


The Transformed Eulerian mean velocities, $\bar{v}^*$ and $\bar{\omega}^*$, are defined by:

$$\bar{v}^* = \bar{v} - \frac{\partial \psi}{\partial p} \text{, the northward component}$$
$$\bar{\omega}^* = \bar{\omega} + \frac{\partial \psi \cos \phi}{a \cos \phi \partial \phi} \text{, the vertical component}$$

The mass stream-function (in units of kg s$^{-1}$), at level $p$, is defined by:

$$\Psi(p) = \frac{2\pi a \cos\phi}{g_0} \left[ \int_p^0 \bar{v} dp - \psi \right]$$

with upper boundary condition (at $p = 0$): $\psi = 0$ and $\Psi = 0$





The eastward wind tendency, $\frac{\partial \bar{u}}{\partial t}|_{\mathrm{adv}(\bar{v}^*)}$ , due to the TEM northward wind advection and Coriolis term
is given by:

$$\frac{\partial \bar{u}}{\partial t}|_{\mathrm{adv}(\bar{v}^*)} = \bar{v}^*[f - \frac{\partial \,\bar{u}\cos\phi}{a\cos\phi\,\partial\phi}]$$


The eastward wind tendency, $\frac{\partial \bar{u}}{\partial t}|_{\mathrm{adv}(\bar{\omega}^*)}$ , due to the TEM vertical wind advection is given by:

$$\frac{\partial \bar{u}}{\partial t}|_{\mathrm{adv}(\bar{\omega}^*)} = \bar{\omega}^*\frac{\partial \bar{u}}{\partial p}$$


***Transformation to log-pressure coordinate***

We define a log-pressure coordinate (Andrews et al 1987) by:

$z = -\mathrm{H}\ln(p/p_0)$ , $\qquad p = p_0 e^{-z/H}$
where: $H = RT_s/g_0$ is a mean scale height of the atmosphere. We recommend to use $H = 7$ km ,
corresponding to $T_s \approx 240$ K , a constant reference air temperature.

The Eliassen-Palm Flux in log-pressure coordinate, $\hat{\mathbf{F}} = \{\hat{F}_{(\phi)}, \hat{F}_{(z)}\}$, is then obtained from the pressure
coordinate form by:

$$\hat{F}_{(\phi)} = \frac{p}{p_0}F_{(\phi)}$$

$$\hat{F}_{(z)} = -\frac{H}{p_0}F_{(p)}$$


The Andrews et al (1987) formulation is then multiplied by the constant reference density $\rho_s =$
$p_0/RT_s$ , which is used in the definition of the background density profile $\rho_0 = \rho_s e^{-z/H}$ in the log-
pressure coordinate system. Here, this scaling is not applied, to maintain the unit of the Eliassen-Palm
flux in $m^3$ $s^{-2}$.

The Eliassen-Palm divergence in log-pressure coordinate is:

$$\boldsymbol{\nabla}_{(z)} \cdot \hat{\mathbf{F}} = \frac{\partial \,\hat{F}_{(\phi)}\cos\phi}{a\cos\phi\,\partial\phi} + \frac{\partial\hat{F}_{(z)}}{\partial z} = \frac{p}{p_0}\boldsymbol{\nabla} \cdot \mathbf{F}$$


The Transformed Eulerian Mean upward wind velocity is:

$$\bar{w}^* = -\frac{H}{p}\bar{\omega}^*$$






***Output***

In summary, the TEM recipe output maps to the CMOR variables listed in Table A1 as follows:
$\hat{F}_{(\phi)} \rightarrow$ epfy, northward component of the Eliassen-Palm Flux
$\hat{F}_{(z)} \rightarrow$ epfz, upward component of the Eliassen-Palm Flux
$\bar{v}^* \rightarrow$ vtem, Transformed Eulerian Mean northward wind
$\bar{w}^* \rightarrow$ wtem, Transformed Eulerian Mean upward wind
$\hat{\Psi} \rightarrow$ psitem, Transformed Eulerian Mean mass stream-function
$\nabla_{(z)} \cdot \hat{\mathbf{F}} \rightarrow$ utendepfd, tendency of eastward wind due to EP Flux divergence
$\frac{\partial \bar{u}}{\partial t}|_{\mathrm{adv}(\bar{v}^*)} \rightarrow$ utendvtem, tendency of eastward wind due to TEM northward wind advection and the
Coriolis term
$\frac{\partial \bar{u}}{\partial t}|_{\mathrm{adv}(\bar{\omega}^*)} \rightarrow$ utendwtem, tendency of eastward wind due to TEM upward wind advection
**Acknowledgements**

DynVarMIP developed from a wide community discussion. We are grateful for the input of many
colleagues. In particular we would like to thank Julio Bachmeister, Thomas Birner, Andrew Charlton-
Perez, Steven Hardiman, Martin Juckes, Alexey Karpechko, Chihirio Kodama, Hauke Schmidt, Tiffany
Shaw, Ayrton Zadra and many others for discussion and their comments on previous versions of the
manuscript or parts of it. EPG acknowledges support from the US National Science Foundation under
grant AGS-1546585.

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





**TABLES**



**Table 1:** Variability. Standard (already in CMIP5) variables at daily and monthly mean frequency. New: more
vertical levels (plev19) for 3D daily and the zonal mean geopotential height, 2D.

| Name | Long name [unit] | Dimension, Grid |
|------|------------------|-----------------|
| psl | Sea Level Pressure [Pa] | 2D, XYT |
| pr | Precipitation [kg m$^{-2}$ s$^{-1}$] | 2D, XYT |
| tas | Near-Surface Air Temperature [K] | 2D, XYT |
| uas | Eastward Near-Surface Wind [m s$^{-1}$] | 2D, XYT |
| vas | Northward Near-Surface Wind [m s$^{-1}$] | 2D, XYT |
| ta | Air Temperature [K] | 3D, XYZT |
| ua | Eastward Wind [m s$^{-1}$] | 3D, XYZT |
| va | Northward Wind [m s$^{-1}$] | 3D, XYZT |
| wap | omega (=dp/dt) [Pa s$^{-1}$] | 3D, XYZT |
| zg | Geopotential Height [m] | 3D, XYZT |
| hus | Specific Humidity [1] | 3D, XYZT |
| zmzg | Geopotential Height [m] | 2D, YZT |



**Table 2:** Momentum (atmosphere). Zonal mean variables (2D, grid: YZT).

| Name (priority) | Long name [unit] | Frequency |
|-----------------|------------------|-----------|
| epfy (1) | northward component of the Eliassen-Palm Flux   [m$^3$ s$^{-2}$] | monthly & daily |
| epfz (1) | upward component of the Eliassen-Palm Flux [m$^3$ s$^{-2}$] | monthly & daily |
| vtem (1) | Transformed Eulerian Mean northward wind [m s$^{-1}$] | monthly & daily |
| wtem (1) | Transformed Eulerian Mean upward wind [m s$^{-1}$] | monthly & daily |
| utendepfd (1) | tendency of eastward wind due to Eliassen-Palm Flux divergence [m s$^{-2}$] | monthly & daily |
| utendnogw (1) | tendency of eastward wind due to nonorographic gravity waves [m s$^{-2}$] | daily |
| utendogw (1) | tendency of eastward wind due to orographic gravity waves [m s$^{-2}$] | daily |
| utendvtem (1) | tendency of eastward wind due to TEM northward wind advection and the Coriolis term  [m s$^{-2}$] | daily |
| utendwtem (1) | tendency of eastward wind due to TEM upward wind advection [m s$^{-2}$] | daily |
| psitem (2) | Transformed Eulerian Mean mass stream-function [kg s$^{-1}$] | daily |



**Table 3.** Momentum (atmosphere). Monthly mean variables (3D, grid: XYZT)

| Name (priority) | Long name [unit] | Frequency |
|-----------------|------------------|-----------|
| utendnogw (1) | tendency of eastward wind due to nonorographic gravity waves [m s$^{-2}$] | monthly |
| utendogw (1) | tendency of eastward wind due to orographic gravity waves [m s$^{-2}$] | monthly |
| vtendnogw (1) | tendency of northward wind due to nonorographic gravity waves [m s$^{-2}$] | monthly |
| vtendogw (1) | tendency of northward wind due to orographic gravity waves [m s$^{-2}$] | monthly |









**Table 4.** Momentum (surface). 2D variables (Grid: XYT)

| Name (priority) | Long name [unit] | Frequency |
|---|---|---|
| tauu (1) | surface downward eastward wind stress [Pa] | daily |
| tauv (1) | surface downward northward wind Stress [Pa] | daily |
| tauupbl (2) | surface downward eastward wind stress due to boundary layer mixing [Pa] | daily |
| tauvpbl (2) | surface downward northward wind stress due to boundary layer mixing [Pa] | daily |

**Table 5.** Heat. 2D zonal mean variables (Grid: YZT)

| Name (priority) | Long name [unit] | Frequency |
|---|---|---|
| zmtnt (1) | tendency of air temperature due to diabatic processes [K s$^{-1}$] | monthly |
| tntrl (1) | tendency of air temperature due to longwave heating [K s$^{-1}$] | monthly |
| tntrs (1) | tendency of air temperature due to shortwave heating [K s$^{-1}$] | monthly |
| tntnogw (2) | tendency of air temperature due to nonorographic gravity wave dissipation [K s$^{-1}$] | monthly |
| tntogw (2) | tendency of air temperature due to orographic gravity wave dissipation [K s$^{-1}$] | monthly |

Note: There is currently duplication in the database for the names of the tendency of air temperature due to longwave / shortwave heating. This is still an open issue. As well, CF standard names might need to be requested for tntnogw and tntogw.

**Table A1.** Momentum budget variable list (2D monthly / daily zonal means, YZT)

| Name | Long name [unit] |
|---|---|
| epfy | northward component of the Eliassen-Palm Flux   [m$^3$ s$^{-2}$] |
| epfz | upward component of the Eliassen-Palm Flux [m$^3$ s$^{-2}$] |
| vtem | Transformed Eulerian Mean northward wind [m s$^{-1}$] |
| wtem | Transformed Eulerian Mean upward wind [m s$^{-1}$] |
| psitem | Transformed Eulerian Mean mass stream-function [kg s$^{-1}$] |
| utendepfd | tendency of eastward wind due to Eliassen-Palm Flux divergence [m s$^{-2}$] |
| utendvtem | tendency of eastward wind due to TEM northward wind advection and the Coriolis term  [m s$^{-2}$] |
| utendwtem | tendency of eastward wind due to TEM upward wind advection [m s$^{-2}$] |

**Table A2.** Input for a TEM diagnostic program (CMOR convention)

| Name | Long name [unit] | Dimension | Frequency |
|---|---|---|---|
| ta | Air temperature [K] | 3D | HF = 6-hour or higher frequency |
| ua | Eastward Wind [m s$^{-1}$] | 3D | HF = 6-hour or higher frequency |
| va | Northward Wind [m s$^{-1}$] | 3D | HF = 6-hour or higher frequency |
| wap | omega (=dp/dt) [Pa s$^{-1}$] | 3D | HF = 6-hour or higher frequency |