# Peer review of "The Dynamics and Variability Model Intercomparison 1 Project (DynVarMIP) for CMIP6: Assessing 2 the Stratosphere-Troposphere System 3"

_Geoscientific Model Development, 2016_

## Short Comment (SC1) · 6 Jun 2016

Dear authors,

In agreement with the CMIP6 panel members, the Executive editors of GMD would like to establish a common naming convention for the titles of the CMIP6 experiment description papers.

The title of CMIP6 papers should include both the acronym of the MIP, and CMIP6, so that it is clear this is a CMIP6-Endorsed MIP.

Good formats for the title include:

[Figure]

'XYZMIP contribution to CMIP6: Name of project'

or

'Name of Project (XYZMIP) contribution to CMIP6'

If you want to include a more descriptive title, the format could be along the lines of,

'XYZMIP contribution to CMIP6: Name of project - descriptive title'

or

'Name of Project (XYZMIP) contribution to CMIP6: descriptive title.'

When you revise your manuscript, please correct the title of your manuscript accordingly.

Additionally, we, the GMD executive editors, strongly recommend to add a version number to the MIP description. The reason for the version numbers is so that the MIP protocol can be updated later, normally in a second short paper outlining the changes. See, for example: http://www.geosci-model-dev.net/special_issue11.html,

Yours,

Astrid Kerkweg

––––––––––––––––––––

---

## Short Comment (SC2) · 13 Jun 2016

Presently, the authors suggest that data for the 4xCO2 simulations be output over the final 40 years of the simulation. However, this will completely miss the initial response of the circulation to the abrupt "turning-on" of CO2 forcing. This initial response is very important for understanding how the troposphere and stratosphere may couple via eddy feedbacks, and provide dynamical mechanisms for the equilibrated response seen in the final years of the simulation. Thus, I would strongly recommend having data output in the first 30 (or 40) years of the 4xCO2 simulation and in the final 30 (or 40 years) of the simulation. This will allow the community to study both the adjustment of the system to the abrupt CO2 forcing, as well as its equilibrated response.

---

## Referee Comment (RC1) · Anonymous Referee #1 · 15 Jun 2016

**A review report on "DynVarMIP: Assessing the Dynamics 1 and Variability of 2 the Stratosphere-Troposphere System" by E. P. Gerber and E. Manzini**

Recommendation: Minor revision

**General comments:**

The paper presents purposes and strategy of DynVarMIP. The importance of the momentum and energy budget of the atmospheric circulation for decreasing uncertainty in projections of future climates including regional climate, precipitation and extreme events responding to natural and anthropogenic forcing is documented. The strategy for the diagnostics is also concretely described. This activity is relevant to WCRP grand challenges mainly on "Clouds, Circulation and Climate Sensitivity", and additionally on "Climate Extremes" and on "Biospheric Forcing and Feedbacks". The description is relatively concise and clear. I think that this paper has a value to be published in Geosci. Model Dev. However, I have minor comments which may make this paper clearer and more easily understood for general readers as well as modelling scientists. Thus, I recommend minor revision before being accepted for publication.

**Comments**

ll. 24-34: The authors mainly emphasized the importance of research on the mid-latitude storm tracks. However, it is also important to examine waves with various scales in various latitudes evenly because all these waves as well as convection and boundary layer processes are interacted with each other and affect the atmospheric circulation. This point should be discussed in more detail.

l. 40: Cumulous convection is also an important parameterized process. This process is related to generation of resolved waves particularly in the tropical region and hence indirectly contribute to the momentum budget of the middle atmosphere. This point should be discussed.

ll. 93-96: A reference is necessary, which describes details of DECK experiment, preindustrial control, abrupt 4x CO2 and 1pctCO2 etc.

l. 291: What CMOR is an abbreviation for?

ll. 313-372: Equation numbers should be added.

ll. 374-385: Equation numbers should be referred to.

l. 518: It is better to add the formulae and/or equation numbers in the table. For example, "tendency of eastward wind due to TEM northward wind advection and the Coriolis term" may have some ambiguity (i.e., $-f\overline{v}^{*}$ or $-f\overline{v}$).

**Others**

l. 209: Remove the second "."

ll. 333 A space is needed after $\nabla \cdot \mathbf{F}$.

---

## Referee Comment (RC2) · Anonymous Referee #2 · 17 Jun 2016

The paper describes overall goals and scopes of the DynVarMIP, one of the diagnostic MIPs of the CMIP6. Objective and scientific questions of the project is concisely described. Proposed diagnostics are also reasonably well defined by listing specific variables of interest in the Appendix.

1. Scientific questions

One of my concerns is that three key questions in section 2 are not well addressed. It would be helpful what the common biases of the current generation of the models, such as CMIP5 models, and why they are important. It is unclear to me what "the role of dynamics in shaping the climate response to anthropogenic forcings" means. Are there any climate responses that are independent of atmospheric dynamics? This question

needs to be better justified. Lastly, it would be helpful to describe what stratospheric processes are important in varying time scales. Since not all readers are familiar with stratosphere-troposphere coupling, one or two paragraph long discussion would be useful. If possible, a simple schematic diagram could be useful here.

2. Link between key questions and diagnostics

It would be useful to relate each diagnostics, briefly outlined in section 3, to three key questions in section 2. To me, all three diagnostics (i.e., variability, momentum, and heat) are focused on the model biases. It is unclear how they are related with questions 2 and 3.

3. Workshop result

It is stated that workshop will be held in June. But, as far as I know, the workshop is already held. It would be helpful what community is concerning about DynVarMIP and what the detailed projects, proposed by DynVar community, for DynVarMIp. These details would be useful for modeler to better understand the nature of the DynVarMIP.

4. Data

Abrupt4*CO2: It is proposed to archive key data for the equilibrium state, year 111-150. But, it would be also interesting to see how circulation reaches equilibrium state by analyzing first 10 or 20 years. Is it possible?

TEM recipe: Please show the mathematical formulation of psitem "on log-p coordinate". I found that utendvtem and utendwtem are computed on pressure coordinate. Is there any reason not to use log-p coordinate?

5. Minor issues:

L100: Please define acronyms of each MIPs. Although this paper is a part of CMIP6 special issue, readers do not need to read all other papers to figure out the acronyms.

L176: Table -> Tables

L181: Shepherd, in 2016 -> Shepherd, 2016

L183: models by in the -> models by the

L189: circulation are -> circulation is

L201: forcing -> forcings

L209: diagnostics.. -> diagnostics. (delete one dot)

L349: add "-" in front of w*

---

## Short Comment (SC3) · 26 Jun 2016

I have a few recommendations to make it clearer to modelling groups exactly how the diagnostics are to be calculated.

(1) line 166 on the calculation of the TEM diagnostics. It is stated that "It is important that these calculations be performed on the native grid of the model (or as close as possible), before being interpolated to standard levels for archival purposes." It's clear in the appendix that these diagnostics are to be calculated on pressure levels, but it's not clear here. In fact it sounds like the calculations should be performed on the native grid, which will not be pressure levels everywhere for most models. I suggest "It is important that these calculations be performed on THE PRESSURE LEVELS AS

CLOSE TO THE NATIVE GRID AS POSSIBLE."

(2) line 278 on the calculation of fluxes. I think it should be emphasized here that the fluxes need to be calculated using instantaneous fields e.g., people may use 6-hourly averages. Suggest "computed from INSTANTANEOUS high frequency data"

(3) line 283 on the calculation of zonal averages. I suggest being more explicit about the proposed best practise for calculating zonal averages. I think it would be much less useful if zonal averages appear as NaNs at any pressure level that intercepts the surface at some point in the longitude circle. For example, that would mean in the Northern Hemisphere, we probably couldn't see the vertical E-P flux below about 600hPa in the mid-latitudes, even though it is a small portion of Asia that is below the surface here. I would suggest either taking a representative zonal average only over the longitudes that are above the surface or performing extrapolation below the ground of the variables that make up the fluxes, using some standard practise e.g. Trenberth, K. E., Berry, J. C. and Buja, L. E. (1993) Vertical Interpolation and Truncation of Model-coordinate Data, NCAR Technical Note NCAR/TN-396+STR, doi:10.5065/D6HX19NH. I expect it is a bit much to ask all modelling groups to perform the latter, so perhaps it is best to ensure that everyone uses a consistent methodology. In which case, the representative zonal average over all longitudes that are above the surface may be the best option.

A second point about the calculation of zonal averages. Line 284, suggests that this can be done online and modelling groups may interpret this as the fluxes etc can be calculated online on their model levels but, unless I'm mistaken, this isn't the way it's supposed to be done. Is it necessary to make the online or offline statements, since this may lead to confusion in that respect?

(4) Table 5. Perhaps this is just a confusion on my part, but is the "tendency of air temperature due to diabatic processes" supposed to be the TOTAL tendency due to diabatic processes i.e., this will be the sum of moist processes, short wave, long wave,

turbulent diffusion, temperature tendencies due to gravity wave drag, tendencies due to any other diffusive processes and energy fixers? If so, then I think would be clearer to call it "Total tendency of temperature due to diabatic processes". It's confusing to have

"tendency of air temperature due to diabatic processes" "tendency of air temperature due to longwave heating" "tendency of air temperature due to shortwave heating"

since longwave heating and shortwave heating are also diabatic processes, so I think added "Total" in front of the first one would be clearer.

I'm also confused about this based on the text at line 196 because it's stated that the set of diagnostics allows us to understand the interaction between radiation, moisture and the circulation. If the first variable in the table is the Total tendency due to diabatic processes, then it's not actually possible to isolate the effect of moisture alone since the total will include other aspects such as turbulent diffusion. This made me wonder whether the first variable in the table is actually supposed to be the "tendency of air temperature due to moist processes"? Either way, I feel like some clarification would be helpful on this variable.
* * *

---

## Short Comment (SC4) · 26 Jun 2016

While providing the TEM velocities will enable a comprehensive analysis of the atmospheric momentum budget, there are differences between the TEM velocities and the Lagrangian transport of mass by the atmospheric circulation. Specifically, the Brewer-Dobson Circulation has historically been deduced both from the residual TEM circulation and from the average time for an air parcel to travel to a given stratospheric sampling region (i.e. the mean age of air or mean age; see chapter 5 of the SPARC CCM-Val2 report). Differences can arise due to isentropic mixing and recirculation (Waugh and Hall 2002, chapter 5 of the CCMVal2 report) - the TEM perspective only contains part of the story.

Furthermore, it has been argued that observational estimates of historical changes in mean age do not agree with the simulated trend towards younger age (Ray et al 2014), while observational estimates of w* must rely on reanalyses products which generally do not agree with one another as to the magnitude and sign of recent trends (Abalos et al 2015) and cannot be used to constrain models. Mean age therefore provides a more sensitive test as to whether model simulated trends in the Brewer-Dobson Circulation are reliable.

I strongly suggest that mean age be archived. Note that only monthly mean and zonal mean mean age is crucial, so the data volume is relatively small.

Abalos, Marta, et al. "Evaluating the advective Brewer-Dobson circulation in three reanalyses for the period 1979–2012." Journal of Geophysical Research: Atmospheres 120.15 (2015): 7534-7554.

Ray, Eric A., et al. "Improving stratospheric transport trend analysis based on SF6 and CO2 measurements." Journal of Geophysical Research: Atmospheres 119.24 (2014).

Waugh, Darryn, and Timothy Hall. "Age of stratospheric air: Theory, observations, and models." Reviews of Geophysics 40.4 (2002).

SPARC, 2010: SPARC CCMVal Report on the Evaluation of Chemistry-Climate Models. V. Eyring, T. Shepherd and D. Waugh (Eds.), SPARC Report No. 5, WCRP-30/2010, WMO/TD - No. 40, available at www.sparc-climate.org/publications/sparc-reports/

---

## Short Comment (SC5) · 29 Jun 2016

These comments reflect the discussion of the breakout discussion group on "The Circulation Response to External Forcing" that we led at the SPARC DynVar Workshop on June 8, 2016.

Kevin Grise and Michael Sigmond

Major Comments:

A chief purpose of DynVarMIP is to provide additional dynamical variables to help understand the mechanisms behind the atmospheric circulation response to external forcing. Yet, the choice of runs and periods of interest for the DynVarMIP output does not

appear to maximize its potential. We have several suggestions that might help to improve the effectiveness of DynVarMIP.

1. DynVarMIP output is requested from the last 40 years of both the abrupt4xCO2 and 1pctCO2 DECK runs. In the abrupt4xCO2 run, this period represents an equilibrated 4xCO2 climate, whereas in the 1pctCO2 run, it represents the transient state of the climate upon CO2 quadrupling (at year 140). However, to really understand the mechanisms behind the atmospheric response to a quadrupling of CO2, it is also essential to study the initial transient response to forcing. A number of papers have provided important insight into the mechanisms behind the atmospheric circulation response by looking at the period immediately after the instantaneous doubling or quadrupling of CO2. For example, Wu et al. (2013) showed that, in the first few months following an instantaneous CO2 doubling, the extratropical circulation response appears to form first at stratospheric levels and then descend into the troposphere. Grise and Polvani (2014a) showed that, in response to an instantaneous quadrupling of CO2, the Southern Hemisphere mid-latitude jet responds faster than the global-mean surface temperature and reaches its equilibrium position within several decades (see their Fig. 10a). Shaw and Voigt (2015) showed that the summertime Pacific jet stream initially shifts poleward during the first 20-30 years after an instantaneous quadrupling of CO2 but then shifts equatorward (as its equilibrium solution would suggest) (see their Fig. 5b). Therefore, given these results, we feel that it is justified to also request DynVarMIP output for the first 40 years of the abrupt4xCO2 run.

2. Notably missing from the list of runs with DynVarMIP output are the CFMIP-led runs: amip4xCO2, amip4K, and amipFuture. These 30-year runs, which are companion runs to the amip runs for which DynVarMIP output is already requested, help to isolate the circulation responses due to radiative forcing only (amip4xCO2) and warming sea surface temperatures only (amip4K and amipFuture). For example, see the recent papers by Grise and Polvani (2014b), Shaw and Voigt (2015), and He and Soden (2015). Given the relatively short length of these runs, we feel that it is justified
to request output from these runs (at least on monthly-mean timescales), given their importance for isolating dynamical mechanisms.

3. A key point of discussion at the recent DynVar Workshop was being able to identify when key circulation responses become distinct from natural variability (a so-called "time of emergence"). It was agreed that this was an important diagnostic to include in future DynVar reports to the broader community. To calculate this diagnostic, it would be necessary to have the DynVarMIP output for all years of the 1pctCO2 run (but only at monthly-mean temporal resolution).

Minor Comments:

1. It would be good to clarify which DAMIP and VolMIP runs include DynVarMIP output. Single forcing runs may be especially important in isolating the mechanisms responsible for the circulation response.

2. Several other variables were discussed in our breakout group that would be useful to include in DynVarMIP output (maybe as priority 2 requests): a. Clear-sky temperature tendencies (on daily timescales) b. Ozone (on daily timescales) c. Potential vorticity (possibly on potential temperature surfaces)

3. It would be valuable to have a certain location (such as a website) to collect metadata that are relevant for understanding the response of the dynamics (e.g., details of orographic and non-orographic gravity wave schemes used, etc.) as such data is not readily documented in the peer-reviewed literature.

References

Grise, K. M., and L. M. Polvani, 2014a: Southern Hemisphere cloud-dynamics biases in CMIP5 models and their implications for climate projections. J. Climate, 27, 6074–6092.

Grise, K. M., and L. M. Polvani (2014b), The response of mid-latitude jets to increased CO2: Distinguishing the roles of sea surface temperature and direct radiative forcing,

Geophys. Res. Lett., 41, 6863–6871.

He, J., and B. J. Soden (2015), Anthropogenic weakening of the tropical circulation: The relative roles of direct CO2 forcing and sea surface temperature change., J. Clim., 28, 8728-8742, doi:10.1175/JCLI-D-15-0205.1.

Shaw, T. A., and A. Voigt (2015), Tug of war on summertime circulation between radiative forcing and sea surface warming, Nature Geosci., 8, 560-566, doi:10.1038/ngeo2449.

Wu, Y., R. Seager, T. A. Shaw, M. Ting, and N. Naik, 2013: Atmospheric circulation response to an instantaneous doubling of carbon dioxide. Part II: Atmospheric transient adjustment and its dynamics. J. Climate, 26, 918-935.
* * *

---

## Short Comment (SC6) · 4 Jul 2016

*Alison Ming and Peter Hitchcock*

**General comments**

We welcome the range of diagnostic variables being requested as part of DynVarMIP especially the diabatic heating rates which will be useful alongside the Transformed Eulerian Mean (TEM) quantities. We believe the ability to study not only the momentum budget but also the mass and thermodynamic budgets will be extremely valuable for the DynVar community (and others). As a general principle, we think it is important that the data requested permit researchers to have a good chance at closing these budgets.

The requested data goes a long way in that direction, but we have a few suggestions for possible improvements.

Our specific comments and suggestions are as follows.

**Specific comments**

Line 95: It may be more valuable to have the first 40 years of the abrupt 4xCO2 runs instead of the final 40 years, if we are also requesting the final 40 years of the 1pctCO2 runs, since we would then have access to both a period of strong transient adjustment as well as a period with an established, strong response.

Line 123: As other comments have also pointed out, it may be useful to recommend what should be done on pressure level grid points which lie below the surface. We would argue that having extrapolated data would be more useful than missing values (following, for instance, the approach given in the NCAR technical note Isla Simpson has referred to), particularly for zonal mean fluxes. If modelling centres do not extrapolate data, representative zonal means for these isobars (e.g., an integral around latitude circles normalized by the fraction of the isobar which lies above the surface) are far better than missing values, but to interpret fluxes correctly one also would need the longitudinally-varying surface pressure (to work out how much of a given isobar lies above the surface at a given latitude and time).

Surface pressure (not sea level pressure) is also essential for the mass budget and for computing the mountain torque. It should be included in Table 1.

Lines 151-160: In addition to the flux and tendency terms in Table 2 and 5, we should request $\overline{u}$ and $\overline{T}$ on the 39-level grid. Having access to the finer scale structure will be essential for investigating dynamical mechanisms, especially near the tropopause. The temperature on a finer grid will be useful for a range of offline radiative calculations.

Moreover, while we appreciate and agree with the importance of the TEM framework, we think the following are good arguments for requesting the more fundamental fluxes

of momentum and heat, $\overline{v'T'}$, $\overline{u'v'}$, $\overline{u'w'}$, rather than the more derived Eliassen-Palm (EP) fluxes.:

(1) Provided that $\overline{u}$ and $\overline{T}$ are available on the higher resolution set of pressure levels (plev39), the EP fluxes can be calculated from these fluxes, subject to computing some vertical and meridional derivatives, and losing some of the temporal covariances on time scales between less than a day (these could be retained in the raw fluxes, but covariances between the raw fluxes and the various derivatives of the mean state would be lost). Spot checks using ERA Interim data do show some errors associated with this interpolation, but they are generally less than 10%; whether this is important of course depends on what a given study has in mind.

(2) Given only the EP fluxes, the reverse is not possible; one cannot recover the raw momentum or heat fluxes. Since many tropospheric studies use an Eulerian framework, having $\overline{u'v'}$ itself will be very useful.

(3) Since they are less derived quantities, there is less risk of the different modeling groups making different decisions about how to compute them, and thus we are more likely to get apples-to-apples comparisons. This is perhaps also an argument for requesting $\overline{v'T'}$ rather than $\overline{v'\theta'}$.

(4) The decision of how to treat pressure levels is simpler for these fluxes – of course missing values are still an issue, but for EP fluxes there are also issues of how to compute vertical derivatives on levels next to those that intersect the surface. Providing the raw fluxes leaves the decision of how to deal with this up to the needs of the study.

(5) Having the heat-fluxes permits calculation of both the Eulerian and Transformed Eulerian Mean meridional circulation.

The main arguments for requesting the EP fluxes are that

(A) The vertical and meridional derivatives can potentially be computed more accurately, on a grid closer to the native model resolution.

(B) Some additional temporal covariances (between the fluxes and the mean state-derivatives) could be retained, if the modeling groups compute the EP fluxes at a higher temporal resolution, then average to daily output.

(C) There would be less work involved for studies interested in the EP fluxes themselves.

However, with the higher vertical resolution of the 39-level grid, it's not clear that (A) is as much of a concern as it is for the coarser 19-level grid, and while there are some issues with sub-daily scale covariances being important in the raw fluxes (e.g. mid-latitude momentum fluxes in the Southern Hemisphere troposphere), we are not aware of major sources of covariance between the fluxes and the mean-state derivatives on these timescales. We would argue that the importance of the Eulerian momentum budget for tropospheric studies and the simpler nature of the request outweighs the potential benefits of (A) and (B). If the vertical gradients are a significant concern, the vertical gradients of zonal wind and temperature could also be requested.

Moreover, if we accept the accuracy of derivatives computed on the 39-level grid and the loss of the temporal covariances on timescales less than a day, the advective tendencies utendvtem and utendwtem can be computed offline, and so do not need to be requested.

For numerical reasons, there can be a difference between the meridional streamfunction calculated from the meridional wind and that calculated from the vertical wind. Integrating the meridional wind seems likely to be the better option though we are not aware of a study demonstrating that fact explicitly. Nonetheless, given any one of the meridional wind, the stream function and the veritical wind, the other two can be computed (again subject to the accuracy of the integration and differentiation). The principle of requesting the simplest variable might therefore suggest that the most sensible course of action is to request $\overline{v}$ as a Priority 1 variable, followed perhaps by $\overline{\omega}$ as a Priority 2 variable. Again, if the heat fluxes and mean temperatures are available,

one can then compute the TEM circulation offline, removing potential inconsistencies in how the heat flux and static stability is treated near the surface.

Finally, in order to close the budget, it would be very useful to have the net tendency of the zonal wind due to all parameterized processes (say, 'utendnet').

We therefore propose replacing the quantities (epfy, epfz, vtem, wtem, utendepfd, utendvtem, utendwtem, psitem) in Table 2 with the following: $\overline{u}$, $\overline{v}$, $\overline{T}$, $\overline{v'T'}$, $\overline{u'v'}$ , $\overline{u'w'}$, and utendnet.

Line 200: Short wave and long wave heating tendencies have a broader applicability in determining circulation changes – they are relevant for determining the mean structure and seasonal cycle of lower stratospheric temperatures [Fueglistaler et al., 2009] , a region with significant biases as shown by Kim et al. (2013) in CMIP5 models and of central importance to stratospheric composition and therefore also to radiative forcing [Forster and Shine, 1997; Solomon et al., 2010; Nowack et al., 2014, Marsh et al., 2016]. They also play a role in determining the structure of the tropical upwelling [Ming et al. 2016a, 2016b].

Lines 203-205: Are these diabatic tendencies due to gravity wave dissipation more important than having a separation of the radiative tendencies into all sky and clear sky? The latter would also be very useful for understanding the diabatic heating budget near the TTL [Wright and Fueglistaler 2013], and could also be requested as Priority 2 fields.

Line 529: In Table 5, is zmtnt the net tendency of all parameterized diabatic processes? This is a bit ambiguous from the text as it stands, but would be the most useful for closing the thermodynamic budget. tntrl and tntrs should be clearly specified as all-sky heating rates.

**References**

Forster, P. M. and K. P. Shine, 2002: Assessing the climate impact of trends in stratospheric water vapor. Geophysical Research Letters, 29 (6), 10–1–10–4.

Fueglistaler, S. et al. (2009) The diabatic heat budget of the upper troposphere and lower/mid stratosphere in ECMWF reanalyses. QJRMS 135:21-37

Kim, J., K. M. Grise, S.-W. Son (2013) Thermal characteristics of the cold-point tropopause region in CMIP5 models. Journal of Geophysical Research: Atmospheres 118, 8827–8841.

Marsh, D., J. -F. Lamarque, A. J. Conley, and L. M. Polvani, 2016: Stratospheric ozone chemistry feedbacks are not critical for the determination of climate sensitivity in CESM1(WACCM). Geophysical Research Letters, 43, 3928-3934, doi:10.1002/2016GL068344.

Ming, A., P. Hitchcock, and P. Haynes, 2016a: The Double Peak in Upwelling and Heating in the Tropical Lower Stratosphere. Journal of the Atmospheric Sciences, 73 (5), 1889–1901, doi:10.1175/JAS-D-15-0293.1.

Ming, A., P. Hitchcock, and P. Haynes, 2016b: The Response of the Lower Stratosphere to Zonally Symmetric Thermal and Mechanical Forcing. Journal of the Atmospheric Sciences, 73 (5), 1903–1922, doi:10.1175/JAS-D-15-0294.1.

Nowack, P. J., Abraham, N. L., Maycock, A. C., Braesicke, P., Gregory, J. M., Joshi, M. M., Osprey, A., et al. (2014). A large ozone-circulation feedback and its implications for global warming assessments. Nature Climate Change, 5 41-45.

Solomon, S., K. H. Rosenlof, R. W. Portmann, J. S. Daniel, S. M. Davis, T. J. Sanford, G-K. Plattner (2010) Contributions of stratospheric water vapor to decadal changes in the rate of global warming. Science 327(5970):1219–1223.

Wright, J. S., and S. Fueglistaler (2013) Large differences in reanalyses of diabatic heating in the tropical upper troposphere and lower stratosphere. Atmos. Chem. Phys. 13, 9565–9576, 2013 doi:10.5194/acp-13-9565-2013.

---

## Author Comment (AC1) · 5 Aug 2016

Thank you for alerting us to this problem with the title. It has been revised to "The Dynamics and Variability Model Intercomparison Project (DynVarMIP) for CMIP6: Assessing the Stratosphere-Troposphere System" to be consistent with the CMIP6 convention.

---

## Author Comment (AC2) · 5 Aug 2016

Reply to SC2: 'output periods for 4xCO2', E.A. Barnes

We very much agree that our initial choice of years would have missed this opportunity to understand the circulation response of the atmosphere to CO2. In the revised manuscript, we now request the first and last 40 years of this integration, to see both the initial response and the final, equilibrated response.

To avoid overtaxing archival requirements, we no longer request any daily data from the 1% per year CO2 integrations.

---

## Author Comment (AC3) · 5 Aug 2016

Reply to SC4: 'Inferring the large scale stratospheric circulation', C. Garfinkel

We agree that the age of air provides valuable – and complementary information – on the stratospheric circulation. The age of air had been on our list of diagnostics in an earlier version of the MIP, but removed upon concern to overburden modeling centers that might not already compute the age. We have added it back to our data request, but at a priority 2 level: it is highly recommended, but not required for participation in the MIP. In the text we propose that the age can be computed with a "clock tracer" and provide references to explain why this is valuable for our understanding of the stratospheric circulation.

---

## Author Comment (AC4) · 5 Aug 2016

Reply to SC5: 'Recommendations for DynVarMIP', K. Grise
(comments in italics, our responses in plain text).

*These comments reflect the discussion of the breakout discussion group on "The Circulation Response to External Forcing" that we led at the SPARC DynVar Workshop on June 8, 2016. Kevin Grise and Michael Sigmond*

We thank you very much for this valuable feedback – and for providing specific references to help justify the MIP requests. We've re-written the paper substantially in light of the DynVar MIP conference.

*Major Comments:*
*A chief purpose of DynVarMIP is to provide additional dynamical variables to help understand the mechanisms behind the atmospheric circulation response to external forcing. Yet, the choice of runs and periods of interest for the DynVarMIP output does not appear to maximize its potential. We have several suggestions that might help to improve the effectiveness of DynVarMIP.*

*1. DynVarMIP output is requested from the last 40 years of both the abrupt4xCO2 and 1pctCO2 DECK runs. In the abrupt4xCO2 run, this period represents an equilibrated 4xCO2 climate, whereas in the 1pctCO2 run, it represents the transient state of the climate upon CO2 quadrupling (at year 140). However, to really understand the mech- anisms behind the atmospheric response to a quadrupling of CO2, it is also essential to study the initial transient response to forcing. A number of papers have provided important insight into the mechanisms behind the atmospheric circulation response by looking at the period immediately after the instantaneous doubling or quadrupling of CO2. For example, Wu et al. (2013) showed that, in the first few months follow- ing an instantaneous CO2 doubling, the extratropical circulation response appears to form first at stratospheric levels and then descend into the troposphere. Grise and Polvani (2014a) showed that, in response to an instantaneous quadrupling of CO2, the Southern Hemisphere mid-latitude jet responds faster than the global-mean surface temperature and reaches its equilibrium position within several decades (see their Fig. 10a). Shaw and Voigt (2015) showed that the summertime Pacific jet stream initially shifts poleward during the first 20-30 years after an instantaneous quadrupling of CO2 but then shifts equatorward (as its equilibrium solution would suggest) (see their Fig. 5b). Therefore, given these results, we feel that it is justified to also request DynVarMIP output for the first 40 years of the abrupt4xCO2 run.*

This issue was brought up by other reviewers as well, and we now request the first and last 40 years of the abrubt 4x CO2 simulation. This choice is justified in the text as well in a new section titled "Experiments" (now the new section 4), which specifically discusses the experiments and their link to our scientific questions.

*2. Notably missing from the list of runs with DynVarMIP output are the CFMIP-led runs: amip4xCO2, amip4K, and amipFuture. These 30-year runs, which are compan- ion runs to the amip runs for which DynVarMIP output is already requested, help to isolate the circulation responses due to radiative forcing only (amip4xCO2) and warming sea surface temperatures only (amip4K and amipFuture). For example, see the recent papers by Grise and Polvani (2014b), Shaw and Voigt (2015), and He and*

*Soden (2015). Given the relatively short length of these runs, we feel that it is justified to request output from these runs (at least on monthly-mean timescales), given their importance for isolating dynamical mechanisms.*

This is a great suggestion, and we've requested DynVar output from these runs. Note, however, that we only require modeling centers to provide output from the DECK experiments to participate in the MIP; this decision was made much earlier, when we sought commitments from the modeling centers to participate. Thus these experiments are "Tier 2" – highly recommend, but not required – and we've sought to justify them to the modeling centers based on the arguments you gave above.

*3. A key point of discussion at the recent DynVar Workshop was being able to identify when key circulation responses become distinct from natural variability (a so-called "time of emergence"). It was agreed that this was an important diagnostic to include in future DynVar reports to the broader community. To calculate this diagnostic, it would be necessary to have the DynVarMIP output for all years of the 1pctCO2 run (but only at monthly-mean temporal resolution).*

Yes. This is now part of our formal request.

*Minor Comments:*
*1. It would be good to clarify which DAMIP and VolMIP runs include DynVarMIP output. Single forcing runs may be especially important in isolating the mechanisms responsible for the circulation response.*

We now provide references to the other MIPs, which clearly spell out which experiments and the diagnostics, and discuss it briefly in the paper. VolMIP request the DynVar diagnostics for their short term volcanic simulations. DAMIP only request the long/short wave heating rates from integrations focused on solar forcing.

*2. Several other variables were discussed in our breakout group that would be useful to include in DynVarMIP output (maybe as priority 2 requests): a. Clear-sky temperature tendencies (on daily timescales) b. Ozone (on daily timescales) c. Potential vorticity (possibly on potential temperature surfaces)*

We've now added a priority 2 request for clear sky temperature tendencies. We feel, however that ozone is outside the scope of our MIP (but it will be carefully consider by the AeroChemMIP). We also feel that potential vorticity (which would be most valuable on potential temperature surfaces) is asking too much. We hope that having the key dynamical variables on a plev19 grid would allow for analysis along these lines.

*3. It would be valuable to have a certain location (such as a website) to collect meta-data that are relevant for understanding the response of the dynamics (e.g., details of orographic and non-orographic gravity wave schemes used, etc.) as such data is not readily documented in the peer-reviewed literature.*

This is a good suggestion to DynVarMIP. We believe it's outside the scope of this paper, but will follow up with it on our webpage.

---

## Author Comment (AC5) · 5 Aug 2016

Reply to SC6: 'Suggestions for clarifying the diagnostic variables being requested.', A. Ming

(comments in italics, our responses in plain text).

*General comments*

*We welcome the range of diagnostic variables being requested as part of DynVarMIP especially the diabatic heating rates which will be useful alongside the Transformed Eulerian Mean (TEM) quantities. We believe the ability to study not only the momentum budget but also the mass and thermodynamic budgets will be extremely valuable for the DynVar community (and others). As a general principle, we think it is important that the data requested permit researchers to have a good chance at closing these budgets.*

*The requested data goes a long way in that direction, but we have a few suggestions for possible improvements.*

We thank you for this very thoughtful set of comments. As explained below, we did not feel it was appropriate to fundamentally reorganize the data request at this late time (given the long discussion we've had with the modeling centers to date), but we believe that you could get these key fluxes with the 3D output that will be made available, either from the DynVarMIP (the 19 level grid) or from other MIPs (for instance HighResMIP) or the standard output, at least at some selected tropospheric levels (the "plev7h" grid), please see our response below.

*Our specific comments and suggestions are as follows.*

*Specific comments*

*Line 95: It may be more valuable to have the first 40 years of the abrupt 4xCO2 runs instead of the final 40 years, if we are also requesting the final 40 years of the 1pctCO2 runs, since we would then have access to both a period of strong transient adjustment as well as a period with an established, strong response.*

This is a good point, echoed by other reviewers. We have changed the manuscript to request the first 40 years of the 4x CO2 simulation (and no longer ask for the last 40 years of the 1pct C02 run, which was a bid redundant with the last 40 years of the 4xCO2 run.)

*Line 123: As other comments have also pointed out, it may be useful to recommend what should be done on pressure level grid points which lie below the surface. We would argue that having extrapolated data would be more useful than missing values (following, for instance, the approach given in the NCAR technical note Isla Simpson has referred to), particularly for zonal mean fluxes. If modelling centres do not ex- trapolate data, representative zonal means for these isobars (e.g., an integral around latitude circles normalized by the fraction of the isobar which lies above the surface) are far better than missing values, but to interpret fluxes correctly one also would need the longitudinally-*

*varying surface pressure (to work out how much of a given isobar lies above the surface at a given latitude and time). Surface pressure (not sea level pressure) is also essential for the mass budget and for computing the mountain torque. It should be included in Table 1.*

As noted in our response to I. Simpson, we do give guidance on this issue. You are correct that surface pressure is also needed, however to correctly interpret the output, and this is now included in our request (Table 1.)

*Lines 151-160: In addition to the flux and tendency terms in Table 2 and 5, we should request u and T on the 39-level grid. Having access to the finer scale structure will be essential for investigating dynamical mechanisms, especially near the tropopause. The temperature on a finer grid will be useful for a range of offline radiative calculations.*

This is a very important suggestion! We agree that zonal mean u and T are needed on the 39 level grid, and have added these to our data requests.

*Moreover, while we appreciate and agree with the importance of the TEM framework, we think the following are good arguments for requesting the more fundamental fluxes of momentum and heat, v0T0, u0v0 , u0w0, rather than the more derived Eliassen-Palm (EP) fluxes.:*
*(1) Provided that u and T are available on the higher resolution set of pressure levels (plev39), the EP fluxes can be calculated from these fluxes, subject to computing some vertical and meridional derivatives, and losing some of the temporal covariances on time scales between less than a day (these could be retained in the raw fluxes, but covariances between the raw fluxes and the various derivatives of the mean state would be lost). Spot checks using ERA Interim data do show some errors associated with this interpolation, but they are generally less than 10%; whether this is important of course depends on what a given study has in mind.*
*(2) Given only the EP fluxes, the reverse is not possible; one cannot recover the raw momentum or heat fluxes. Since many tropospheric studies use an Eulerian frame- work, having u0v0 itself will be very useful.*
*(3) Since they are less derived quantities, there is less risk of the different modeling groups making different decisions about how to compute them, and thus we are more likely to get apples-to-apples comparisons. This is perhaps also an argument for re-questing v0T 0 rather than v0✓0.*
*(4) The decision of how to treat pressure levels is simpler for these fluxes – of course missing values are still an issue, but for EP fluxes there are also issues of how to compute vertical derivatives on levels next to those that intersect the surface. Providing the raw fluxes leaves the decision of how to deal with this up to the needs of the study.*
*(5) Having the heat-fluxes permits calculation of both the Eulerian and Transformed Eulerian Mean meridional circulation.*
*The main arguments for requesting the EP fluxes are that*
*(A) The vertical and meridional derivatives can potentially be computed more accu-rately, on a grid closer to the native model resolution.*

*(B) Some additional temporal covariances (between the fluxes and the mean state-derivatives) could be retained, if the modeling groups compute the EP fluxes at a higher temporal resolution, then average to daily output.*

*(C) There would be less work involved for studies interested in the EP fluxes them- selves. However, with the higher vertical resolution of the 39-level grid, it's not clear that (A) is as much of a concern as it is for the coarser 19-level grid, and while there are some issues with sub-daily scale covariances being important in the raw fluxes (e.g. mid-latitude momentum fluxes in the Southern Hemisphere troposphere), we are not aware of major sources of covariance between the fluxes and the mean-state derivatives on these timescales. We would argue that the importance of the Eulerian momentum budget for tropospheric studies and the simpler nature of the request outweighs the potential benefits of (A) and (B). If the vertical gradients are a significant concern, the vertical gradients of zonal wind and temperature could also be requested.*

*Moreover, if we accept the accuracy of derivatives computed on the 39-level grid and the loss of the temporal covariances on timescales less than a day, the advective ten- dencies utendvtem and utendwtem can be computed offline, and so do not need to be requested.*

*For numerical reasons, there can be a difference between the meridional streamfunc- tion calculated from the meridional wind and that calculated from the vertical wind. Integrating the meridional wind seems likely to be the better option though we are not aware of a study demonstrating that fact explicitly. Nonetheless, given any one of the meridional wind, the stream function and the veritical wind, the other two can be computed (again subject to the accuracy of the integration and differentiation). The principle of requesting the simplest variable might therefore suggest that the most sen- sible course of action is to request v as a Priority 1 variable, followed perhaps by ! as a Priority 2 variable. Again, if the heat fluxes and mean temperatures are available,*

*one can then compute the TEM circulation offline, removing potential inconsistencies in how the heat flux and static stability is treated near the surface.*

*Finally, in order to close the budget, it would be very useful to have the net tendency of the zonal wind due to all parameterized processes (say, 'utendnet').*

*We therefore propose replacing the quantities (epfy, epfz, vtem, wtem, utendepfd, utendvtem, utendwtem, psitem) in Table 2 with the following: u, v, T, v0T0, u0v0 , u0w0, and utendnet.*

The decision to request the raw fluxes (e.g. u'v', u'w' and v'T') as opposed to the more derived E-P fluxes was one that we debated ourselves earlier in the set up of this MIP. One of the initial ideas was to ask for u'v', v'T' etc., and then to additionally ask for d \Theta / d p (or equivalently, d T / dp); this allows one to accurately compute the E-P fluxes. The divergence of the E-P flux, however, involves a second vertical derivative, and so effectively a second vertical derivative of T. (And with coarse data, results are sensitive to how these derivatives are calculated) So it is still not clear to us, which vertical resolution could be recommended.

It is true that the 39 level output is at relatively high vertical resolution. However, please note that in the troposphere, the 39 and 19 level grids are nearly identical, and that half of the additional levels of the 39 level grid simply cover the mesosphere, hence these levels

will be there only for a few set of models. For direct comparison, here we report the two grids:

plev19: 1000, 925, 850, 700, 600, 500, 400, 300, 250, 200, 150, 100, 70, 50, 30, 20, 10, 5 and 1 hPa

plev39: 1000 925 850 700 600 500 400 300 250 200 170 150 130 115 100 90 80 70 50 30 20 15 10 7 5 3 2 1.5 1 0.7 0.5 0.4 0.3 0.2 0.15 0.1 0.07 0.05 0.03 hPa

Therefore, we think that the 19 level grid (the DynVarMIP proposed gird for 3-D daily data) will allow for a reasonable analysis of the heat and momentum fluxes in the CMIP6 models, in the tropospheric mid-latitudes as well as for a relative range of levels in the stratosphere. In addition, from the workshop we perceived considerable interest in separating the impact of planetary scale Rossby waves from synoptic scale waves, which won't be possible with zonal mean TEM diagnostics, but it will be possible using this new 3D data set. While in the case of higher frequency output, the fluxes can be computed, at some selected levels, from other data requests (HighResMIP or standard output), see for instance the plev7h grid (925, 850, 700, 600, 500, 250, 50 hPa).

Please also note, that originally we planned to ask all DynVarMIP output on the plev19 grid (or maybe a 23 level grid with a few additional levels in the stratosphere). The 39 level grid was later proposed, but it's still not entirely clear to us whether it will be adopted by all modeling centers.

After considering all of the above, we think that it is best to remain with our proposition to ask for the TEM diagnostics; we can therefore provide the community with more advanced (in the sense of highly derived) – albeit standard – diagnostics. The TEM diagnostics are indeed nowadays used as standard tools to evaluate models and are asked as well by HighResMIP and AeroChemMIP.

*Line 200: Short wave and long wave heating tendencies have a broader applicability in determining circulation changes – they are relevant for determining the mean structure and seasonal cycle of lower stratospheric temperatures [Fueglistaler et al., 2009] , a region with significant biases as shown by Kim et al. (2013) in CMIP5 models and of central importance to stratospheric composition and therefore also to radiative forcing [Forster and Shine, 1997; Solomon et al., 2010; Nowack et al., 2014, Marsh et al., 2016]. They also play a role in determining the structure of the tropical upwelling [Ming et al. 2016a, 2016b].*

This is a good point, and we've revised the text (citing Fueglistaler et al. 2009 and Kim et al. 2013) to make a stronger argument for this decomposition.

*Lines 203-205: Are these diabatic tendencies due to gravity wave dissipation more important than having a separation of the radiative tendencies into all sky and clear sky? The latter would also be very useful for understanding the diabatic heating budget near the TTL [Wright and Fueglistaler 2013], and could also be requested as Priority 2 fields.*

Gravity waves are a particular focus of DynVarMIP (and coordinated with the HiResMIP), but we agree that this decomposition would be very valuable. We now request the clear sky only radiative temperature tendency as well (but as a priority 2 variable, as not to overtax the modeling centers).

*Line 529: In Table 5, is zmtnt the net tendency of all parameterized diabatic processes? This is a bit ambiguous from the text as it stands, but would be the most useful for closing the thermodynamic budget. tntrl and tntrs should be clearly specified as all-sky heating rates.*

We agree that this was unclear, and we've adjusted the text and table to clarify that we are asking for the total tendency from all processes, and that tntrl and tntrs are all-sky heating rates. In addition, we have requested the temperature tendencies associated with moist processes alone (at priority 2), to help untangle the contribution of different processes to the total heat budget.

---

## Author Comment (AC6) · 5 Aug 2016

Reply to SC3: 'Recommendations for clarifications on the calculation of some of the diagnostics listed', I. Simpson

We thank Isla Simpson for these valuable comments on the manuscript, and respond in line below (I. Simpson comments in italics, our responses in plain text).

*(1) line 166 on the calculation of the TEM diagnostics. It is stated that "It is important that these calculations be performed on the native grid of the model (or as close as possible), before being interpolated to standard levels for archival purposes." It's clear in the appendix that these diagnostics are to be calculated on pressure levels, but it's not clear here. In fact it sounds like the calculations should be performed on the native grid, which will not be pressure levels everywhere for most models. I suggest "It is important that these calculations be performed on THE PRESSURE LEVELS AS CLOSE TO THE NATIVE GRID AS POSSIBLE."*

Yes, this is important, and we follow your advice to make this explicit, both within the main text and within the appendix.

*(2) line 278 on the calculation of fluxes. I think it should be emphasized here that the fluxes need to be calculated using instantaneous fields e.g., people may use 6-hourly averages. Suggest "computed from INSTANTANEOUS high frequency data"*

Yes – we made the suggested change.

*(3) line 283 on the calculation of zonal averages. I suggest being more explicit about the proposed best practise for calculating zonal averages. I think it would be much less useful if zonal averages appear as NaNs at any pressure level that intercepts the surface at some point in the longitude circle. For example, that would mean in the Northern Hemisphere, we probably couldn't see the vertical E-P flux below about 600hPa in the mid-latitudes, even though it is a small portion of Asia that is below the surface here. I would suggest either taking a representative zonal average only over the longitudes that are above the surface or performing extrapolation below the ground of the variables that make up the fluxes, using some standard practise e.g. Trenberth, K. E., Berry, J. C. and Buja, L. E. (1993) Vertical Interpolation and Truncation of Model- coordinate Data, NCAR Technical Note NCAR/TN-396+STR, doi:10.5065/D6HX19NH. I expect it is a bit much to ask all modelling groups to perform the latter, so perhaps it is best to ensure that everyone uses a consistent methodology. In which case, the representative zonal average over all longitudes that are above the surface may be the best option.*

This is a very important point that we had overlooked. We now recommend extrapolation to avoid any missing data, but also allow for modeling centers to take representative zonal (and time) averages in cases where data is missing from subset of the zonal (time) points. We suggest that if more than half of the data is missing, however, then it would be best to report the data as NaN.

*A second point about the calculation of zonal averages. Line 284, suggests that this can be done online and modelling groups may interpret this as the fluxes etc can be calculated online on their model levels but, unless I'm mistaken, this isn't the way it's*

*supposed to be done. Is it necessary to make the online or offline statements, since this may lead to confusion in that respect?*

We meant that an online calculation might be possible for models that use a pressure coordinate, but we agree this was overly complicating the situation. We've now removed the discussion of online/offline calculations, and simply ask that they be done on instantaneous data sample at least 4 times a day.

*(4) Table 5. Perhaps this is just a confusion on my part, but is the "tendency of air temperature due to diabatic processes" supposed to be the TOTAL tendency due to diabatic processes i.e., this will be the sum of moist processes, short wave, long wave, turbulent diffusion, temperature tendencies due to gravity wave drag, tendencies due to any other diffusive processes and energy fixers? If so, then I think would be clearer to call it "Total tendency of temperature due to diabatic processes". It's confusing to have "tendency of air temperature due to diabatic processes" "tendency of air temperature due to longwave heating" "tendency of air temperature due to shortwave heating"since longwave heating and shortwave heating are also diabatic processes, so I think added "Total" in front of the first one would be clearer.*

We have clarified this in the text and table to clarify that we are asking for the total tendency from all processes.

*I'm also confused about this based on the text at line 196 because it's stated that the set of diagnostics allows us to understand the interaction between radiation, moisture and the circulation. If the first variable in the table is the Total tendency due to diabatic processes, then it's not actually possible to isolate the effect of moisture alone since the total will include other aspects such as turbulent diffusion. This made me wonder whether the first variable in the table is actually supposed to be the "tendency of air temperature due to moist processes"? Either way, I feel like some clarification would be helpful on this variable.*

We agree that the scientific logic was unclear here. In response to another comment, be do believe that the total diabatic tendency is important, as it will allow for a careful budget analysis. But to understand the impact of moisture, it would be valuable to quantify the impact of moist processes alone. We have therefore requested this (but at priority 2, as not to overtax modeling centers who initially agreed to the total tendency).

---

## Author Response (AR1)

**Reply to RC1: 'A review report', Anonymous Referee #1**
(review in italics, our responses in plain text).

*The paper presents purposes and strategy of DynVarMIP. The importance of the momentum and energy budget of the atmospheric circulation for decreasing uncertainty in projections of future climates including regional climate, precipitation and extreme events responding to natural and anthropogenic forcing is documented. The strategy for the diagnostics is also concretely described. This activity is relevant to WCRP grand challenges mainly on "Clouds, Circulation and Climate Sensitivity", and additionally on "Climate Extremes" and on "Biospheric Forcing and Feedbacks". The description is relatively concise and clear. I think that this paper has a value to be published in Geosci. Model Dev. However, I have minor comments which may make this paper clearer and more easily understood for general readers as well as modelling scientists. Thus, I recommend minor revision before being accepted for publication.*

We thank the review for this careful review, and believe that the manuscript has improved in response to these concerns and suggestions.

***Comments***
*ll. 24-34: The authors mainly emphasized the importance of research on the mid-latitude storm tracks. However, it is also important to examine waves with various scales in various latitudes evenly because all these waves as well as convection and boundary layer processes are interacted with each other and affect the atmospheric circulation. This point should be discussed in more detail.*

The initial emphasis on storm tracks was done to link more closely with the Grand Challenge on Clouds, Circulation, and Sensitivity. But we certainly did not mean to limit our selves to this region alone. We've added a new sentence to this paragraph emphasizing the global nature of any regional circulation problem. The sentence reads:

"Wave coupling between the tropics and high latitudes (e.g. Li et al., 2015) make regional circulation change a global problem, requiring a careful assessment of dynamical processes across all latitudes."

In addition, in response to the second reviewer, we've provided a very brief review of the importance of stratosphere-troposphere interactions in weather, which better emphasizes the global nature of the research interest of the DynVarMIP. The new paragraph reads:

"The stratosphere impacts tropospheric weather (e.g. though blocking events; Anstey et al., 2013; Shaw et al., 2014), and an improved representation of stratospheric processes can improve synoptic weather forecasts (e.g. Gerber et al., 2012; McTaggart-Cowan et al., 2011). Coupling between the stratospheric polar vortices and the tropospheric jet streams enhances subseasonal and seasonal predictability in the midlatitudes (e.g. Baldwin and Dunkerton, 2001; Roff et al., 2011; Sigmond et al., 2013), while in the tropics, the Quasi-Biennial Oscillation affects subseasonal variability and preciptiation (e.g. Yoo and Son, 2016) and provides a source of enhanced interannual predictability (e.g. Boer and Hamilton, 2008). The stratosphere has also been implicated in the ENSO teleconnections to the extratropics (e.g. Bell et al., 2009; Cagnazzo and Manzini 2009) and linked with decadal variability in the Atlantic (e.g. Reichler et al., 2012). Finally, the stratosphere plays an important role in climate change (e.g. Scaife et al. 2011), particularly through ozone loss and recovery over Antarctica (e.g. Gerber and Son, 2014; Min and Son, 2013; Thompson et al., 2011; Wilcox and Charlton-Perez, 2013) and through changes in stratospheric water vapor, which impact surface temperatures and climate sensitivity (e.g. Dessler et al., 2013; Solomon et al., 2010)."

*l. 40: Cumulous convection is also an important parameterized process. This process is related to generation of resolved waves particularly in the tropical region and hence indirectly contribute to the momentum budget of the middle atmosphere. This point should be discussed.*

We've included a reference to parameterized convective processes here, and discuss it in more detail in section 3.2. The reviewer is correct to note that there are additional parameterized processes that affect the momentum budget of the free troposphere, and the cumulative effect of these processes will be estimated as a residual in the momentum budget. The new paragraph in section 3.2 reads:

"Additional parameterized processes can impact momentum transport in the free atmosphere, including convective momentum transport, vertical diffusion, and sponge layers near the model top (often used to prevent artificial wave reflection). Numerical diffusion can also artificially impact the momentum transport. The impact of these processes will be diagnosed in aggregate, however, as a residual between the total momentum tendency by the resolved flow and gravity waves and the actual change in the resolved flow."

*ll. 93-96: A reference is necessary, which describes details of DECK experiment, preindustrial control, abrupt 4x CO2 and 1pctCO2 etc.*

We've included a new section ('4. Experiments' in the revised paper) that discusses the experiments in more detail and includes all the necessary references.

*l. 291: What CMOR is an abbreviation for?*

Climate Model Output Rewriter – this is now stated in the manuscript.

*ll. 313-372: Equation numbers should be added.*
*ll. 374-385: Equation numbers should be referred to.*
*l. 518: It is better to add the formulae and/or equation numbers in the table. For example, "tendency of eastward wind due to TEM northward wind advection and the Coriolis term" may have some ambiguity (i.e or ).*

Done. We agree and we have added the equation numbers, and refer to them in the mapping (at lines376-385 of the original manuscript).

*l. 209: Remove the second ".".*
*ll.333A space is needed after · .*
Done.

**Reply to RC2: 'Review of "DynVarMIP: Assessing the Dynamics and Variability of the Stratosphere-Troposphere System" by Gerber and Manzini', Anonymous Referee #2**
(review in italics, our responses in plain text)

*The paper describes overall goals and scopes of the DynVarMIP, one of the diagnostic MIPs of the CMIP6. Objective and scientific questions of the project is concisely described. Proposed diagnostics are also reasonably well defined by listing specific variables of interest in the Appendix.*

We thank the reviewer for this careful review, and believe that the manuscript has improved significantly as a result of our efforts to acknowledge these concerns.

*1. Scientific questions*
*One of my concerns is that three key questions in section 2 are not well addressed. It would be helpful what the common biases of the current generation of the models, such as CMIP5 models, and why they are important. It is unclear to me what "the role of dynamics in shaping the climate response to anthropogenic forcings" means. Are there any climate responses that are independent of atmospheric dynamics? This question needs to be better justified. Lastly, it would be helpful to describe what stratospheric processes are important in varying time scales. Since not all readers are familiar with stratosphere-troposphere coupling, one or two paragraph long discussion would be useful. If possible, a simple schematic diagram could be useful here.*

We've made several changes in response to this overall concern about the connection between the diagnostics and research questions.

First, we now explicit stated that climate models have a problem with the storm tracks, particularly in the austral hemisphere. In both CMIP3 and CMIP5 models, it is biased equatorward and too persistent. References are also provided. The new sentence reads:

"Accurate simulation of the storm track climatology and variability has long proved a challenge for climate prediction models, particularly in the austral hemisphere, where the storm track and associated midlatitude jet stream is generally located too far equatorward and is too persistent (e.g. Kidston and Gerber, 2010; Simpson and Polvani, 2016; Swart and Fyfe, 2012, Wenzel et al., 2016)."

Second, we've added a new section ('4. Experiments' in the revised document) that discusses the experiments for which the DynVar diagnostics are requested, and relates them to the three scientific questions. It includes references to papers that analyzed biases in the CMIP models, such as Wenzel et al. 2016 that documented the equatorward bias of the austral jet stream in CMIP5 models, and linked it to biases in future projections.

Third, we've expanded our introduction to include a brief summary of stratosphere-troposphere interactions, providing a number of references for the interested reader (see response to Reviewer 1). As recent review papers, such as Kidston et al. 2015, have included schematic diagrams, we felt it might not be necessary for this paper which is focused more on the technical details of the DynVarMIP.

Lastly, we agree that our second question was written too vaguely – to the point of being vacuous – and have sharpened it: "What is the role of atmospheric heat and momentum transport in shaping the climate response to anthropogenic forcings", to emphasize the connection with our diagnostics.

*2. Link between key questions and diagnostics*
*It would be useful to relate each diagnostics, briefly outlined in section 3, to three key questions in section 2. To me, all three diagnostics (i.e., variability, momentum, and heat) are focused on the model biases. It is unclear how they are related with questions 2 and 3.*

As noted above, a new section has been added to the paper (4 in the revised paper), which discusses the experiments and their relation to the scientific questions in more detail. We've also provided a number of references to studies, which have linked dynamical mechanisms to CMIP5 models. For example, the initial response of atmosphere in the 4xCO2 experiment (as suggested by the reviewer below) is an excellent opportunity to focus on question 2.

*3. Workshop result*
*It is stated that workshop will be held in June. But, as far as I know, the workshop is already held. It would be helpful what community is concerning about DynVarMIP and what the detailed projects, proposed by DynVar community, for DynVarMIp. These details would be useful for modeler to better understand the nature of the DynVarMIP.*

This section has been fully rewritten in light of the results of the workshop. Three groups have been organized to focus on the three science questions of the MIP.

*4. Data*
*Abrupt4\*CO2: It is proposed to archive key data for the equilibrium state, year 111-150. But, it would be also interesting to see how circulation reaches equilibrium state by analyzing first 10 or 20 years. Is it possible?*

This is a good suggestion, and was echoed by many at the DynVarMIP organization meeting in June. We've now requested the first 40 years of this simulation, in addition to the last 40 years, which provide an opportunity to understand the equilibrated response of the model.

*TEM recipe: Please show the mathematical formulation of psitem "on log-p coordinate". I found that utendvtem and utendwtem are computed on pressure coordinate. Is there any reason not to use log-p coordinate?*

We recommend pressure coordinates because most models are in p-coordinate and for models in geometric-z coordinate, prior to the diagnostic calculation it is necessary to transform the input variables to pressure coordinates (Hardiman et al. 2010) to avoid spurious differences. Climate models are usually not written in log-p coordinate.

utendvtem has the same formulation, in both coordinates.
utendwtem does not require a transformation.

We need to keep psitem in kg s-1, to keep it consistent with CMIP already existing conventions; hence we have not given the formulation of psitem in log-p.

*5. Minor issues:*
*L100: Please define acronyms of each MIPs. Although this paper is a part of CMIP6 special issue, readers do not need to read all other papers to figure out the acronyms.*

This is now done (and we provide the up-to-date references as well.)

*L176: Table -> Tables*
*L181: Shepherd, in 2016 -> Shepherd, 2016 L183: models by in the -> models by the*
*L189: circulation are -> circulation is*
*L201: forcing -> forcings*
*L209: diagnostics.. -> diagnostics. (delete one dot)*
*L349: add "-" in front of w\**

Done – thank you for spotting these mistakes.

**The Dynamics and Variability Model Intercomparison Project (DynVarMIP) for CMIP6: Assessing the Stratosphere-Troposphere System**

Edwin P. Gerber[1] and Elisa Manzini[2]

[1] Courant Institute of Mathematical Sciences, New York University, 251 Mercer Street, New York NY 10012, USA.

[2] Max-Planck-Institut für Meteorologie, Bundesstraße 53, 20146 Hamburg, Germany

*Correspondence to:* Elisa Manzini (elisa.manzini@mpimet.mpg.de)

**Abstract.** Diagnostics of atmospheric momentum and energy transport are needed to investigate the origin of circulation biases in climate models and to understand the atmospheric response to natural and anthropogenic forcing. Model biases in atmospheric dynamics are one of the factors that increase uncertainty in projections of regional climate, precipitation, and extreme events. Here we define requirements for diagnosing the atmospheric circulation and variability across temporal scales and for evaluating the transport of mass, momentum and energy by dynamical processes in the context of the Coupled Model Intercomparison Project Phase 6 (CMIP6). These diagnostics target the assessments of both resolved and parameterized dynamical processes in climate models, a novelty for CMIP, and are particularly vital for assessing the impact of the stratosphere on surface climate change.

**Keywords:** Atmosphere, dynamics, momentum and energy transfer, variability, climate and climate change.

**1. Introduction**

The importance and challenge of addressing the atmospheric circulation response to global warming have recently been highlighted by Shepherd (2014) and Vallis et al. (2015). Understanding circulation changes in the atmosphere, particularly of the mid-latitude storm tracks, has been identified by the World Climate Research Programme (WCRP) as one of the grand challenges in climate research. Accurate simulation of the storm track climatology and variability has long proved a challenge for climate prediction models, particularly in the austral hemisphere, where the storm track and associated mid-latitude jet stream is generally located too far equatorward and is too persistent (e.g. Kidston and Gerber, 2010; Simpson and Polvani, 2016; Swart and Fyfe, 2012, Wenzel et al., 2016). The storm tracks depend critically on the transport of momentum, heat and chemical constituents throughout the whole atmosphere. Changes in the storm tracks are thus significantly coupled with lower atmosphere processes such as planetary boundary layer, surface temperature gradients and moisture availability (e.g. Garfinkel et al., 2011; Booth et al., 2013), as well as with processes in the stratosphere, from natural variability on synoptic to intraseasonal timescales (e.g. Baldwin and Dunkerton, 2001) to the response to changes in stratospheric ozone (e.g. Son et al., 2008) and other anthropogenic forcings (e.g. Scaife et al., 2012).  Wave coupling between the tropics and high latitudes (e.g. Li et al., 2015) makes regional circulation change a global problem, requiring a careful assessment of dynamical processes across all latitudes.

The Dynamics and Variability Model Intercomparison Project (DynVarMIP) is an endorsed participant in the Coupled Model Intercomparison Project Phase 6 (CMIP6).  Rather then proposing new experiments, the DynVarMIP requests additional model output from existing CMIP6 experiments. This additional output is critical for understanding the role of atmospheric dynamics in past, present and future climate. Both resolved processes (e.g. Rossby waves, large scale condensation) and parameterized processes (e.g. gravity waves, subgrid-scale convection, and the planetary boundary layer) play important roles in the dynamics and circulation of the atmosphere in models.  DynVarMIP seeks to ensure that sufficient diagnostics of key processes in climate models are archived. Without this model output, we will not be able to fully assess the dynamics of mass, momentum, and heat transport - essential ingredients in projected circulation changes - nor take advantage of the increasingly accurate representation of the stratosphere in coupled climate models. Our rational is that by simply extending the standard output relative to that in CMIP5 for a selected set of experiments, there is potential for significantly expanding our research capabilities in atmospheric dynamics.

Investigation of the impact of solar variability and volcanic eruptions on climate also relies heavily on atmospheric wave forcing diagnostics, as well as radiative heating rates (particularly in the short wave). By extending our request to the energy budget and including diagnostics such as diabatic heating from cloud-precipitation processes, research on the links between moist processes and atmospheric dynamics will be enabled as well. The interplay between moist processes and circulation is central to the WCRP Grand Challenge on Clouds, Circulation and Climate Sensitivity (Bony et al., 2015).

The CMIP5 saw a significant upward expansion of models with a more fully resolved stratosphere (e.g. Gerber et al., 2012), and several multi-model studies have investigated the role of the stratosphere in present climate and in projections of future climate (e.g. Charlton-Perez et al., 2013; Lott et al., 2014; Manzini et al., 2014). The stratosphere impacts tropospheric weather (e.g. though blocking events; Anstey et al., 2013; Shaw et al., 2014), and an improved representation of stratospheric processes can improve synoptic weather forecasts (e.g. Gerber et al., 2012; McTaggart-Cowan et al., 2011). Coupling between the stratospheric polar vortices and the tropospheric jet streams enhances subseasonal and seasonal predictability in the midlatitudes (e.g. Baldwin and Dunkerton, 2001; Roff et al., 2011; Sigmond et al., 2013), while in the tropics, the Quasi-Biennial Oscillation affects subseasonal variability and preciptiation (e.g. Yoo and Son, 2016) and provides a source of enhanced interannual predictability (e.g. Boer and Hamilton, 2008). The stratosphere has also been implicated in the ENSO teleconnections to the extratropics (e.g. Bell et al., 2009; Cagnazzo and Manzini 2009) and linked with decadal variability in the Atlantic (e.g. Reichler et al., 2012).  Finally, the stratosphere plays an important role in climate change (e.g. Scaife et al., 2012), particularly through ozone loss and recovery

Edwin Gerber 22/7/2016 10:59

Edwin Gerber 21/7/2016 14:27

Edwin Gerber 21/7/2016 14:28

Edwin Gerber 22/7/2016 10:59

Edwin Gerber 21/7/2016 14:30

Edwin Gerber 22/7/2016 09:53

Edwin Gerber 21/7/2016 14:31

over Antarctica (e.g. Gerber and Son, 2014; Min and Son, 2013; Thompson et al., 2011; Wilcox and Charlton-Perez, 2013) and through changes in stratospheric water vapor, which impact surface temperatures and climate sensitivity (e.g. Dessler et al., 2013; Solomon et al., 2010). These studies document a growing interest in the role of middle and upper atmosphere in climate (cf. Kidston et al., 2015). New research in this direction will take full advantage of the DynVarMIP diagnostics.

**2. Objectives and Scientific Questions**

The DynVarMIP focuses on the interactions between atmospheric variability, dynamics and climate change, with a particular emphasis on the two-way coupling between the troposphere and the stratosphere. To organize the scientific activity within the MIP, we have identified the following key questions:

1. How do dynamical processes contribute to persistent model biases in the mean state and variability of the atmosphere, including biases in the position, strength, and statistics of the storm tracks, blocking events, and the stratospheric polar vortex?
2. What is the role of atmospheric momentum and heat transport in shaping the climate response to anthropogenic forcings (e.g. global warming, ozone depletion) and how do dynamical processes contribute to uncertainty in future climate projections and prediction?
3. How does the stratosphere affect climate variability at intra-seasonal, inter-annual and decadal time scales?

Investigation of these topics will allow the scientific community to address the role of atmospheric dynamics in the key CMIP6 science questions concerning the origin and consequences of systematic model biases, the response of the Earth System to forcing, and how to assess climate change given climate variability (Eyring et al., 2016). In particular, there is a targeted effort to contribute to the storm track theme of the Clouds, Circulation and Climate Sensitivity Grand Challenge. The DynVarMIP focus on daily fields and diagnostics of the atmospheric flow is also relevant to the Grand Challenge on Climate Extremes, and could also enable contributions to the additional theme on Biospheric Forcings and Feedbacks.

**3. The Diagnostics**

The DynVarMIP requests both enhanced archival of standard variables from the CMIP5 and new diagnostics to enable analysis of both resolved and parameterized processes relevant to the dynamics of the atmosphere. The diagnostics are organized around three scientific themes, as detailed below.

**3.1 Atmospheric variability across scales (short name: *variability*)**
* * *
Edwin Gerber 22/7/2016 10:31

Unknown

Edwin Gerber 22/7/2016 11:23

Edwin Gerber 20/7/2016 09:08

Edwin Gerber 5/8/2016 10:20
**Comment [1]:** A table here more clearly spelling out the years would be helpful here.

Edwin Gerber 5/8/2016 10:20
**Comment [2]:** Did we actually get into these other MIPs? I have only carefully tracked VolMIP, where we are explicitly mentioned in the paper.

Edwin Gerber 21/7/2016 12:21

The first request of the DynVarMIP is enhanced archival of standard variables (listed in Table 1) as daily and monthly means. While modeling centers have been archiving increasingly fine horizontal resolution (close to the native model grid), vertical sampling has been limited to standard levels that changed little from CMIP3 to 5.

The need for enhanced vertical resolution is particularly acute in the upper troposphere and lower stratosphere (UTLS), where there are steep vertical gradients in dynamical variables (e.g. temperature and wind) and chemical constituents (e.g. water vapor and ozone) across the tropopause. Without this finer vertical resolution, analyses of the UTLS would be limited by vertical truncation errors, preventing us from taking full advantage of increased horizontal resolution offered in new model integrations.

A number of other MIPs, in particular the HighResMIP (High Resolution Model Intercomparison Project, Haarsma et al., 2016), have also recognized the need for enhanced vertical resolution for daily data. A common proposed request, the "plev19" grid of pressure levels, has consequently been reached (Martin Juckes, personal communication, see: https://earthsystemcog.org/site_media/projects/wip/CMIP6_pressure_levels.pdf). The pressure levels of the plev19 grid are 1000, 925, 850, 700, 600, 500, 400, 300, 250, 200, 150, 100, 70, 50, 30, 20, 10, 5, and 1 hPa.

The diagnostics in Table 1 will allow for evaluation of atmospheric variability across time and spacial scales, e.g. the assessment of model biases in blocking events, the tropospheric storm tracks, and the stratospheric polar vortices. Comparison between the pre-industrial control, historical, and idealized integrations will allow for evaluation of the response of atmospheric variability to external forcings.

Novel to CMIP6 is also the daily zonal mean geopotential (zmzg, Table 1), tailored to the need of DCPP (Decadal Climate Prediction Project, Boer et al., 2016) to analyze variability on longer time scales and for a large number experiments, while minimizing storage requirements.

**3.2 Atmospheric zonal momentum transport (short name: *momentum*)**

The second group of diagnostics focuses on the transport and exchange of momentum within the atmosphere and between the atmosphere and surface. These diagnostics are listed in Tables 2, 3 and 4. Within this group, a number of new (to CMIP) diagnostics and variables are requested. The goal of this set is to properly evaluate the role of both the resolved circulation and the parameterized dynamical processes in momentum transport. As daily timescales must be archived to capture the role of synoptic processes, we focus on the zonal mean circulation, thereby greatly reducing the total output that must be stored permanently. We have also prioritized the new variables, as noted in Tables 2, 3 and 4. Priority 1 variables are essential to the MIP and required for participation. Priority 2 variables would be very valuable to the MIP, but not are necessary for participation.

Edwin Gerber 2/8/2016 11:49

Elisa Manzini 5/8/2016 16:52

Elisa Manzini 5/8/2016 17:08

The zonal mean quantities (for both daily and monthly means) are requested on the "plev39" grid of pressure levels:  1000, 925, 850, 700, 600, 500, 400, 300, 250, 200, 170, 150, 130, 115, 100, 90, 80, 70, 50, 30, 20, 15, 10, 7, 5, 3, 2, 1.5, 1, 0.7, 0.5, 0.4, 0.3, 0.2, 0.15, 0.1, 0.07, 0.05, and 0.03 hPa. This sampling will allow for detailed exploration of the vertical momentum transport, from the surface to the mesosphere. Subsampling is allowed for models with lower vertical resolution or lower model tops. All three dimensional fields, however, are requested on the plev19 grid.

Models largely resolve the planetary and synoptic scale processes that dominate the transport of momentum within the free atmosphere. Quantification of this transport, however, depends critically on vertical and horizontal wave propagation.  The Transformed Eulerian Mean (TEM) framework allows one to efficiently quantify this momentum transport by waves, in addition to estimating the Lagrangian transport of mass by the circulation (e.g. Andrews and McIntyre, 1976; 1978).  In the stratosphere, the TEM circulation is thus far more relevant to transport of trace gases (e.g. ozone and water vapor) than the standard Eulerian mean circulation (e.g. Butchart 2014). We have therefore request diagnostics based on the TEM framework (see Table 2). The details of these calculations are presented in the Appendix, and further insight can be found in the textbooks by Andrews et al., (1987; pages 127-130) and Vallis (2006; chapter 12).

As seen in the Appendix, the TEM diagnostics depend critically on the vertical structure of the circulation, i.e. vertical derivatives of basic atmospheric state and of wave fluxes, and can only be accurately computed from instantaneous fields, as opposed to daily means. Even with the enhanced "plev39" vertical resolution requested above for the standard meteorological variables, we would not be able to reproduce these statistics from the archived output. It is therefore important that these calculations be performed on pressure levels as close to the native grid of the model as possible, before being interpolated to standard levels for archival purposes.

Dynamical processes, which need to be parameterized because they are not resolved on the grid of the model, also play an important role in momentum transport. Gravity waves transport momentum from the surface to the upper troposphere and beyond, but cannot be properly resolved at conventional climate models resolution. Their wave stresses play a key role in the large scale circulation of the troposphere (e.g. the storm tracks; Palmer et al., 1986) and are primary driver of the stratospheric circulation (e.g. Alexander et al., 2010, and references therein). Atmospheric circulation changes have been shown to be sensitive to the parameterization of gravity waves (e.g., Sigmond and Scinocca, 2010).  The availability of tendencies from gravity wave processes (Tables 2 and 3) will enable a systematic evaluation of this driving term of the circulation, so far largely unexplored in a multi-model context.

Additional parameterized processes can impact momentum transport in the free atmosphere, including convective momentum transport, vertical diffusion, and sponge layers near the model top (often used to prevent artificial wave reflection).  Numerical diffusion can also artificially impact the momentum transport. The impact of these processes will be diagnosed in aggregate, however, as a residual between the total momentum tendency by the resolved flow and gravity waves and the actual change in the resolved flow.

While the TEM circulation approximates the Lagrangian transport of mass, trace gases with sinks and sources in the stratosphere, such as ozone, are also strongly affected by quasi-horizontal mixing along isentropic surfaces (e.g. Plumb, 2002). Breaking Rossby waves rearrange mass along isentropic surfaces: this yields no net movement of mass, but a trace gas with horizontal gradient experiences a net transport. The "age of air" can be used to assess the impact of this mixing, and provides complementary information to the TEM for the assessment of the stratospheric circulation (e.g. Waugh and Hall, 2002). The age can be quantified by a so-called "clock tracer," a passive tracer with a unit source near the surface; the age is then simply the difference between the concentration at the surface and other points in the atmosphere. This variable is requested at priority 2: not required for participation, but requested from models that have this capability.

Diagnostics to archive the parameterized surface stresses are listed in Table 4. A number of studies have documented that the large scale circulation and storm track structure are sensitive to the surface drag (e.g. Chen et al. 2007; Garfinkel et al. 2011; Polichtchouk and Shepherd 2016). These diagnostics will also allow us to connect the CMIP6 with the investigation of weather prediction models by the Working Group on Numerical Experimentation (WGNE) Drag Project (http://collaboration.cmc.ec.gc.ca/science/rpn/drag_project/). To understand how models arrive at the total surface stress, we also request the component due to turbulent processes, usually parameterized by the planetary boundary layer (PBL) scheme, including those stresses that come from subgrid orographic roughness elements. The role of other processes could then be diagnosed by residual.

Evaluation of the resolved and parameterized processes that effect the circulation is essential to diagnosing and understanding model biases in the mean state and variability of the atmosphere, and for diagnosing the processes driving circulation changes in response to natural and anthropogenic forcing. A careful dynamic analysis of circulation change is a critical step in developing a fundamental understanding of the underlying mechanisms, and hence for improving confidence in future projections. We need to know that models not only agree in the response, but that they agree for the same reasons.

**3.3 The atmospheric heat budget (short name: *heat*)**

This set of diagnostics allows us to understand the interaction between radiation, moisture, and the circulation. As with our momentum diagnostics, we request only zonal mean statistics, to limit the additional storage load (Table 5). We ask for the temperature tendency due to all parameterized physics (e.g. all diabatic processes: radiation, convection, boundary layer, stratiform condensation/evaporation, vertical diffusion, etc). Temperature tendencies due to resolved dynamics and numerical diffusion not associated with parameterized physics are then diagnosed in aggregate, as

Edwin Gerber 20/7/2016 08:53

Edwin Gerber 20/7/2016 08:54

Edwin Gerber 3/8/2016 14:42

Edwin Gerber 25/7/2016 10:48

Edwin Gerber 25/7/2016 10:49

Edwin Gerber 25/7/2016 10:52

a residual between the temperature tendency due to all diabatic processes and the actual change in the resolved temperature. To separate the contribution of radiative transfer, we ask for the temperature tendencies due to longwave / shortwave radiative transfer (all sky). If available, the tendencies due to nonorographic / orographic gravity wave dissipation, due to convection (all parameterized types), due to stratiform clouds and precipitation (all type of resolved, large scale clouds and precipitation) and the tendencies due to clear sky longwave / shortwave radiative transfer are requested at priority 2. These would allow for a more careful assessment of dynamical, radiative, moisture and cloud processes on the diabatic heat budget (e.g. Wright and Fuegistaler, 2013; Ming et al., 2016).

Separately diagnosing the short and long wave heating tendencies has proven to be useful for interpreting circulation changes in general (e.g. Fuegistaler et al., 2009; Kim et al., 2013), and is particularly important for understanding the role of solar and volcanic forcings on the circulation. It will allow us to separate the direct impact of changes in solar radiation and aerosol loading from the atmospheric response to these perturbations, and enable analysis to break down feedbacks in Earth System models.

**4. Experiments**

The DynVar diagnostics are requested from the Diagnostic, Evaluation, and Characterization of Klima (DECK) experiments and CMIP6 historical simulations (Eyring et al. 2016) and a total of four closely related experiments; one experiment from the Scenario Model Intercomparison Project (ScenarioMIP; O'Neill et al. 2016) and three experiments from the Cloud Feedback Model Intercomparison Project (CFMIP; Webb et al. 2016), as listed in Table 6. To limit the total data storage, the diagnostics are requested for targeted 40-year periods (detailed in Table 6), with the exception of the *1% yr$^{-1}$ CO2 concentration increase* experiment from the DECK, where only monthly mean diagnostics are requested. As indicated by the third column of Table 6, diagnostics from the DECK and CMIP6 historical simulation are required for participation in the DynVarMIP. Diagnostics from the experiments organized by ScenarioMIP and CFMIP are optional, but highly recommended for modeling centers that participate in these MIPs.

Diagnostics from the *pre-industrial control, AMIP*, and CMIP6 historical simulations are most relevant to our first scientific objective, to understand biases in atmospheric circulation and variability. In particular, the circulation in the latter two experiments can be directly compared against atmospheric reanalyses of the observed atmosphere. Comparison against integrations under strong anthropogenic influence (the last 40 years of the *abrupt quadroupling of CO2* experiment and years 2061-2100 from the *SSP5-RCP8.5* experiment) will help reveal how biases in the historical climatology related to biases in the future climate projections (e.g. Wenzel et al. 2016).

Our second objective is to understand the circulation response to anthropogenic forcing, and will be served by analysis of the equilibrated response of the atmosphere to 4xCO2 and the late 21$^{st}$ century
* * *
**Elisa Manzini 7/8/2016 17:32**

**Edwin Gerber 2/8/2016 12:30**

**Edwin Gerber 2/8/2016 12:30**

**Elisa Manzini 7/8/2016 11:30**

circulation in the *SSP5-RCP8.5* experiment. Wu et al. (2013), Grise and Polvani (2014a), and Shaw and Voigt (2015), however, have shown how the initial response of the atmosphere to an abrupt quadroupling of CO2 reveals a great deal about the dynamical mechanism(s) and their associated time scales; hence our request for the first 40 years of this integration. A number of studies from the CMIP5 have also demonstrated the utility of AMIP climate change experiments, the *amip-p4K*, *amip-future4K*, and *amip-4xCO2* organized by the CFMIP, in isolating the mechanisms for circulation changes (e.g. Grise and Polvani, 2014b; He and Soden, 2015; Shaw and Voigt, 2015). We have therefore requested diagnostics from these simulations from modeling centers, which are also participating in the CFMIP.

Lastly, diagnostics are requested from the full 150 year record from the *1 % yr$^{-1}$ CO2 concentration increase* experiment, specifically to determine the time of emergence in circulation changes. To limit the cost of archiving this data, only monthly mean fields are requested.

Our final objective, to understand the role of stratosphere in surface climate and variability, will be served by a number of these simulations. The *pre-industrial control* and final 40 years of the *abrupt quadroupling of CO2* integrations, however, will be particularly ideal for understanding the role of stratosphere in natural, unforced variability in past and future climates, respectively.

The DynVar diagnostics (or relevant subsets thereof) have been coordinated with diagnostic requests of other CMIP6 endorsed MIPs. The TEM and stratospheric circulation diagnostics are highly relevant to integrations with ozone depleting substances in the Aerosols and Chemistry (AeroChemMIP; Collins et al. 2016) and to the short term response of the atmosphere to volcanic forcing, as detailed in the Volcanic Forcings Model Intercomparison Project (VolMIP; Zanchettin et al. 2016). The zonal mean long and short wave heating rates have been requested for integrations focused on solar variability in the Detection and Attribution MIP (DAMIP; Gillett et al. 2016). Zonal mean geopotential height has been requested as part of the Decadal Climate Prediction Project (DCPP; Boer et al. 2016). Finally, the enhanced archival of daily data and gravity wave drag diagnostics were coordinated with the High Resolution Model Intercomparison Project (HighResMIP; Haarsma et al. 2016).

**5. Analysis Plan**

The DynVarMIP has been organized in response to our experience in coordinating community based, collaborative analysis of coupled climate models from the CMIP5 through the SPARC DynVar activity (e.g. Gerber et al., 2012; Charlton-Perez et al., 2013; Manzini et al., 2014). An analysis plan for the MIP was formulated at an open workshop held in Helsinki, Finland in June 2016. The workshop was attended by approximately 70 scientists from around the world, with broad representation from the modeling and research communities, and held jointly with a subset of the SPARC Reanalysis Intercomparison Project (S-RIP). Three groups were organized to coordinate analysis of the DynVarMIP research objectives.

The first group focused on model biases, and will begin with a systematic analysis of the TEM

circulation and momentum budget in CMIP6 models. A community paper (or potentially a series of papers) is being organized to follow up more systematically on Hardiman et al. 2013, which compared the residual circulation across a subset of CMIP5 models where the relevant diagnostics could be collected on an ad hoc basis. The first paper will focus the momentum and heat balances of the historical climate, where it can be directly compared with observations. Several of the group members are involved in the S-RIP chapter on the Brewer-Dobson Circulation, bringing expertise on potential limitations in our understanding of the momentum and heat budgets in reanalysis.

Two approaches were suggested for the DynVarMIP objective on the response of the circulation to anthropogenic forcing. The first is to extend the systematic, community organized analysis of the heat and momentum budgets to climate change scenarios, with an emphasis on links between models'

ability to capture the past climate with their projections of future circulation changes. The second is to continue informal coordination of research on the underlying mechanisms. Based on past experience, we have found that research on a mechanistic understanding of the atmosphere is often best organized organically, rather than from a top down approach. The potential for a review paper on model hierarchies, which help link basic research to comprehensive climate models, was raised, and will be explored in greater detail at the upcoming WCRP workshop on model hierarchies in November, 2016.

A third group focused on the natural variability of the atmosphere, with a particular emphasis on initial condition predictability (i.e. predictability of the first kind; Lorenz, 1975) in CMIP6 models across a range of time scales, from synoptic to decadal. Charlton et al. (2013) concluded that a better representation of the stratosphere in climate models strongly impacts the variability of the stratosphere, and it is an open question as to the extent which this improves the representation of natural variability in the troposphere. Subseasonal variability was identified as an important, but less explored area in climate research. It is also a time scale for which the stratosphere is particularly relevant, and a review paper was proposed to motivate more systematic analysis of variability on this time scale in CMIP6

models.

To ensure continued participation and collaboration with the modeling centers, representatives from the modeling centers have been invited to participate in the scientific analysis and papers. A future workshop (tentatively set for 2019 at which time CMIP6 data is expected to be available) will be arranged to ensure completion of the analysis.

**6. Conclusions and Outlook**

The goal of the DynVarMIP is to evaluate and understand the role of dynamics in climate model biases and in the response of the climate system to external forcing. This goal is motivated by the fact that

Edwin Gerber 25/7/2016 11:14

We have found that research on a mechanistic understanding of the atmosphere and on rectifying model biases is often best organized organically, rather than from a top down approach. The TEM diagnostics, for example, have been used in a number of CMIP5 studies (e.g. Hardiman et al., 2013; Manzini et al., 2014), but had to be assembled on an ad hoc basis with a limited number models. DynVarMIP is seeking to expand this research by making the key diagnostics available to all. ¶

Edwin Gerber 22/7/2016 12:07

Edwin Gerber 21/7/2016 13:22

[revised manuscript text omitted]

Eulerian mean velocities, $\overline{v}^*$ and $\overline{\omega}^*$, the mass stream-function, $\Psi$.

The Eliassen-Palm flux is a 2-dimesional vector, $\mathbf{F} = \{F_{(\phi)}, F_{(p)}\}$, with northward and vertical components respectively defined by:

$$F_{(\phi)} = a \cos \phi \left\{ \frac{\partial \overline{u}}{\partial p} \psi - \overline{u'v'} \right\} \qquad (2)$$

$$F_{(p)} = a \cos \phi \left\{ \left[ f - \frac{\partial \overline{u} \cos \phi}{a \cos \phi \partial \phi} \right] \psi - \overline{u'\omega'} \right\} \qquad (3)$$

where:

$$\psi = \overline{v'\theta'} / \frac{\partial \overline{\theta}}{\partial p} \qquad (4)$$

Elisa Manzini 4/8/2016 15:19

Elisa Manzini 4/8/2016 15:19

is the eddy stream-function.

The Eliassen-Palm divergence, $\mathbf{\nabla} \cdot \mathbf{F}$, is defined by:

$$\mathbf{\nabla} \cdot \mathbf{F} = \frac{\partial\, F_{(\phi)} \cos\phi}{a \cos\phi \partial\phi} + \frac{\partial F_{(p)}}{\partial p} \qquad (5)$$

The Transformed Eulerian mean northward and vertical velocities are respectively defined by:

$$\overline{v}^* = \overline{v} - \frac{\partial\psi}{\partial p} \qquad\qquad (6)$$

$$\overline{\omega}^* = \overline{\omega} + \frac{\partial\psi\cos\phi}{a\cos\phi\partial\phi} \qquad\qquad (7)$$

The mass stream-function (in units of kg s$^{-1}$), at level $p$, is defined by:

$$\Psi(p) = \frac{2\pi a \cos\phi}{g_0} \left[\int_p^0 \overline{v}\,dp - \psi\right] \qquad (8)$$

with upper boundary condition (at $p = 0$): $\psi = 0$ and $\Psi = 0$

The eastward wind tendency, $\frac{\partial\overline{u}}{\partial t}|_{\mathrm{adv}(\overline{v}^*)}$, due to the TEM northward wind advection and Coriolis term is given by:

$$\frac{\partial\overline{u}}{\partial t}|_{\mathrm{adv}(\overline{v}^*)} = \overline{v}^*\left[f - \frac{\partial\overline{u}\cos\phi}{a\cos\phi\partial\phi}\right] \qquad (9)$$

The eastward wind tendency, $\frac{\partial\overline{u}}{\partial t}|_{\mathrm{adv}(\overline{\omega}^*)}$, due to the TEM vertical wind advection is given by:

$$\frac{\partial\overline{u}}{\partial t}|_{\mathrm{adv}(\overline{\omega}^*)} = -\overline{\omega}^*\frac{\partial\overline{u}}{\partial p} \qquad (10)$$

***Transformation to log-pressure coordinate***

We define a log-pressure coordinate (Andrews et al 1987) by:

$$z = -H \ln(p/p_0) \qquad (11)$$

$$p = p_0 e^{-z/H} \qquad (12)$$

where: $H = RT_s/g_0$ is a mean scale height of the atmosphere. We recommend to use $H = 7$ km, corresponding to $T_s \approx 240$ K, a constant reference air temperature.

The Eliassen-Palm Flux in log-pressure coordinate, $\widehat{\mathbf{F}} = \{\widehat{F}_{(\phi)}, \widehat{F}_{(z)}\}$, is then obtained from the pressure coordinate by:

$$\widehat{F}_{(\phi)} = \frac{p}{p_0} F_{(\phi)} \qquad (13)$$

$$\widehat{F}_{(z)} = -\frac{H}{p_0} F_{(p)} \qquad (14)$$

Elisa Manzini 7/8/2016 11:41

Elisa Manzini 4/8/2016 15:26

Elisa Manzini 4/8/2016 15:26

Elisa Manzini 4/8/2016 15:47

The Andrews et al (1987) formulation is then multiplied by the constant reference density $\rho_s =$

$p_0/RT_s$, which is used in the definition of the background density profile $\rho_0 = \rho_s e^{-z/H}$ in the log- pressure coordinate system. Here, this scaling is not applied, to maintain the unit of the Eliassen-Palm flux in $m^3\ s^{-2}$.

The Eliassen-Palm divergence in log-pressure coordinate is:

$$\mathbf{\nabla}_{(z)} \cdot \widehat{\mathbf{F}} = \frac{\partial \widehat{F}_{(\phi)} \cos\phi}{a \cos\phi \partial\phi} + \frac{\partial \widehat{F}_{(z)}}{\partial z} = \frac{p}{p_0} \mathbf{\nabla} \cdot \mathbf{F} \qquad (15)$$

The Transformed Eulerian Mean upward wind velocity is:

$$\overline{w}^* = -\frac{H}{p}\overline{\omega}^* \qquad (16)$$

***Output***

In summary, the TEM recipe output maps to the CMOR variables listed in Table A1 as follows:

$\widehat{F}_{(\phi)} \rightarrow$ epfy, northward component of the Eliassen-Palm Flux, Eq. (13)

$\widehat{F}_{(z)} \rightarrow$ epfz, upward component of the Eliassen-Palm Flux, Eq. (14)

$\overline{v}^* \rightarrow$ vtem, Transformed Eulerian Mean northward wind, Eq. (6)

$\overline{w}^* \rightarrow$ wtem, Transformed Eulerian Mean upward wind, Eq. (16)

$\widehat{\Psi} \rightarrow$ psitem, Transformed Eulerian Mean mass stream-function, Eq. (8)

$\mathbf{\nabla}_{(z)} \cdot \widehat{\mathbf{F}} \rightarrow$ utendepfd, tendency of eastward wind due to EP Flux divergence, Eq. (15)

$\frac{\partial \overline{u}}{\partial t}\big|_{\text{adv}(\overline{v}^*)} \rightarrow$ utendvtem, tendency of eastward wind due to TEM northward wind advection and the

Coriolis term, Eq. (9)

$\frac{\partial \overline{u}}{\partial t}\big|_{\text{adv}(\overline{\omega}^*)} \rightarrow$ utendwtem, tendency of eastward wind due to TEM upward wind advection, Eq. (10)

**Acknowledgements**

DynVarMIP developed from a wide community discussion. We are grateful for the input of many colleagues. In particular we would like to thank Julio Bachmeister, Thomas Birner, Andrew Charlton-

Perez, Steven Hardiman, Martin Juckes, Alexey Karpechko, Chihirio Kodama, Hauke Schmidt, Tiffany

Shaw, Ayrton Zadra and many others for discussion and their comments on previous versions of the manuscript or parts of it. We gratefully acknowledge the insights and comments from the Reviewers and the interactive Commenters. Their remarks, together with the lively discussions and presentations at the DynVar workshop in Helsinki, have significantly improved the manuscript. We extend our thanks to Alexey Karpechko for his smooth running of the workshop in Helsinki. EPG acknowledges support from the US National Science Foundation under grant AGS-1546585.

**TABLES**
**Table 1:** Variability. Standard (already in CMIP5) variables at daily and monthly mean frequency. New: more
vertical levels (plev19) for 3D daily and the zonal mean geopotential height, 2D.

| Name | Long name [unit] | Dimension, Grid |
|---|---|---|
| psl | Sea Level Pressure [Pa] | 2D, XYT |
| ps | Surface Air Pressure [Pa] | 2D, XYT |
| pr | Precipitation [kg m$^{-2}$ s$^{-1}$] | 2D, XYT |
| tas | Near-Surface Air Temperature [K] | 2D, XYT |
| uas | Eastward Near-Surface Wind [m s$^{-1}$] | 2D, XYT |
| vas | Northward Near-Surface Wind [m s$^{-1}$] | 2D, XYT |
| ta | Air Temperature [K] | 3D, XYZT |
| ua | Eastward Wind [m s$^{-1}$] | 3D, XYZT |
| va | Northward Wind [m s$^{-1}$] | 3D, XYZT |
| wap | omega (=dp/dt) [Pa s$^{-1}$] | 3D, XYZT |
| zg | Geopotential Height [m] | 3D, XYZT |
| hus | Specific Humidity [1] | 3D, XYZT |
| zmzg | Geopotential Height [m] | 2D, YZT |

**Table 2:** Momentum (atmosphere). Zonal mean variables (2D, grid: YZT) on the plev39 grid. The zonal mean
zonal wind is requested, as it would otherwise be unavailable at this vertical resolution.

| Name (priority) | Long name [unit] | Frequency |
|---|---|---|
| ua (1) | eastward wind [m s$^{-1}$] | monthly & daily |
| epfy (1) | northward component of the Eliassen-Palm Flux [m$^3$ s$^{-2}$] | monthly & daily |
| epfz (1) | upward component of the Eliassen-Palm Flux [m$^3$ s$^{-2}$] | monthly & daily |
| vtem (1) | Transformed Eulerian Mean northward wind [m s$^{-1}$] | monthly & daily |
| wtem (1) | Transformed Eulerian Mean upward wind [m s$^{-1}$] | monthly & daily |
| utendepfd (1) | tendency of eastward wind due to Eliassen-Palm Flux divergence [m s$^{-2}$] | monthly & daily |
| utendnogw (1) | tendency of eastward wind due to nonorographic gravity waves [m s$^{-2}$] | daily |
| utendogw (1) | tendency of eastward wind due to orographic gravity waves [m s$^{-2}$] | daily |
| utendvtem (1) | tendency of eastward wind due to TEM northward wind advection and the Coriolis term [m s$^{-2}$] | daily |
| utendwtem (1) | tendency of eastward wind due to TEM upward wind advection [m s$^{-2}$] | daily |
| psitem (2) | Transformed Eulerian Mean mass stream-function [kg s$^{-1}$] | daily |
| mnstrage (2) | mean age of stratospheric air [yr] | monthly |

**Table 3.** Momentum (atmosphere). Monthly mean variables (3D, grid: XYZT) on the plev19 grid.

| Name (priority) | Long name [unit] | Frequency |
|---|---|---|
| utendnogw (1) | tendency of eastward wind due to nonorographic gravity waves [m s$^{-2}$] | monthly |
| utendogw (1) | tendency of eastward wind due to orographic gravity waves [m s$^{-2}$] | monthly |
| vtendnogw (1) | tendency of northward wind due to nonorographic gravity waves [m s$^{-2}$] | monthly |
| vtendogw (1) | tendency of northward wind due to orographic gravity waves [m s$^{-2}$] | monthly |

**Table 4.** Momentum (surface). 2D variables (Grid: XYT)

| Name (priority) | Long name [unit] | Frequency |
|---|---|---|
| tauu (1) | surface downward eastward wind stress [Pa] | daily |
| tauv (1) | surface downward northward wind Stress [Pa] | daily |
| tauupbl (2) | surface downward eastward wind stress due to boundary layer mixing [Pa] | daily |
| tauvpbl (2) | surface downward northward wind stress due to boundary layer mixing [Pa] | daily |

**Table 5.** Heat (atmosphere). 2D zonal mean variables (2D grid: YZT) on the plev39 grid. The zonal mean
temperature is requested, as it would otherwise be unavailable at this vertical resolution.

| Name (priority) | Long name [unit] | Frequency |
|---|---|---|
| ta (1) | air temperature [K] | monthly |
| tntmp (1) | tendency of air temperature due to model physics [K s$^{-1}$] | monthly |
| tntrl (1) | tendency of air temperature due to longwave heating, all sky [K s$^{-1}$] | monthly |
| tntrs (1) | tendency of air temperature due to shortwave heating, all sky [K s$^{-1}$] | monthly |
| tntrlcs (2) | tendency of air temperature due to longwave heating, clear sky [K s$^{-1}$] | monthly |
| tntrscs (2) | tendency of air temperature due to shortwave heating, clear sky [K s$^{-1}$] | monthly |
| tntc (2) | tendency of air temperature due to convection [K s$^{-1}$] | monthly |
| tntscp (2) | tendency of air temperature due to stratiform clouds and precipitation [K s$^{-1}$] | monthly |
| tntnogw (2) | tendency of air temperature due to nonorographic gravity wave dissipation [K s$^{-1}$] | monthly |
| tntogw (2) | tendency of air temperature due to orographic gravity wave dissipation [K s$^{-1}$] | monthly |

**Table 6.** Experiments and integration years for which the DynVarMIP diagnostics are requested.

| Experiment | Collection Period(s) | Tier |
|---|---|---|
| **DECK** (Eyring et al., 2016) | | |
| AMIP | 1979-2014 (ideally for 3 ensemble members) | 1 |
| Pre-industrial control | 111-150 years after the branching point | 1 |
| Abrupt quadrupling of CO2 concentration | years 1-40 and 111-150 | 1 |
| 1 % yr$^{-1}$ CO2 concentration increase | years 1-150 (**monthly mean data only**) | 1 |
| **CMIP6 historical simulation** | | |
| Past ~ 1.5 centuries | 1961-2000 | 1 |
| **ScenarioMIP** (O'Neill et al., 2016) | | |
| SSP5-RCP8.5 | 2061-2100 | 2 |
| **CFMIP** (Webb et al., 2016) | | |
| amip-p4K | 1979-2014 | 2 |
| amip-future4K | 1979-2014 | 2 |
| amip-4xCO2 | 1979-2014 | 2 |

**Comments (margin):**

Elisa Manzini 7/8/2016 11:42

Elisa Manzini 5/8/2016 14:08

Elisa Manzini 5/8/2016 14:09

Elisa Manzini 5/8/2016 18:15

**Table A1.** Momentum budget variable list (2D monthly / daily zonal means, YZT)

| Name | Long name [unit] |
|------|------------------|
| epfy | northward component of the Eliassen-Palm Flux $[m^3\ s^{-2}]$ |
| epfz | upward component of the Eliassen-Palm Flux $[m^3\ s^{-2}]$ |
| vtem | Transformed Eulerian Mean northward wind $[m\ s^{-1}]$ |
| wtem | Transformed Eulerian Mean upward wind $[m\ s^{-1}]$ |
| psitem | Transformed Eulerian Mean mass stream-function $[kg\ s^{-1}]$ |
| utendepfd | tendency of eastward wind due to Eliassen-Palm Flux divergence $[m\ s^{-2}]$ |
| utendvtem | tendency of eastward wind due to TEM northward wind advection and the Coriolis term $[m\ s^{-2}]$ |
| utendwtem | tendency of eastward wind due to TEM upward wind advection $[m\ s^{-2}]$ |

**Table A2.** Input for a TEM diagnostic program (CMOR convention)

| Name | Long name [unit] | Dimension | Frequency |
|------|------------------|-----------|-----------|
| ta | Air temperature [K] | 3D | HF = 6-hour or higher frequency |
| ua | Eastward Wind $[m\ s^{-1}]$ | 3D | HF = 6-hour or higher frequency |
| va | Northward Wind $[m\ s^{-1}]$ | 3D | HF = 6-hour or higher frequency |
| wap | omega (=dp/dt) $[Pa\ s^{-1}]$ | 3D | HF = 6-hour or higher frequency |